# Therapeutic hypothermia for neonatal hypoxic ischaemic encephalopathy in Sub-Saharan Africa: A scoping review

Naa A. Buxton-Tetteh[1], Shakti Pillay[1], Gugulabatembunamahlubi T. J. Kali[2], Alan R. Horn[1]*

**1** Department of Paediatrics and Child Health, Division of Neonatal Medicine, University of Cape Town, Cape Town, South Africa, **2** Department of Paediatrics, Stellenbosch University, Cape Town, South Africa

* alan.horn@uct.ac.za

## Abstract

### Introduction

There are divergent views and limited data regarding therapeutic hypothermia (TH) for neonatal hypoxic ischaemic encephalopathy (HIE) in sub-Saharan Africa (SSA). Our aim was to map and synthesize the published literature describing the use of TH for HIE in SSA, and the associated outcomes.

### Method

We searched Pubmed, Scopus, Google Scholar, and Web of Science from 1 January 1996 to 31 December 2023 for research studies, protocols, feasibility studies and surveys on term and near-term babies with HIE (population) treated with TH (concept) in SSA (context).

### Results

Thirty records were included: Three surveys, one feasibility study and 26 publications describing 23 studies of 21 cohorts, cooling 1420 babies in South Africa, Uganda, and Ghana. There were five studies recruiting at follow-up, five pilot studies, one randomised controlled trial, one case series, and 10 birth cohorts. The methods and design of the studies were highly variable and often inadequate. Only three studies with adequately described and validated cooling methods, non-selective sequential recruitment, and neurological outcomes were identified. Two studies of babies from birth, both with intensive care facilities, reported survival with normal/mildly abnormal outcome in 71% at discharge in one study, and 71% at 12 months in another, with 16% cerebral palsy (CP) in survivors, and only 16% loss to follow-up. The third study, which only included clinic attenders after TH without intensive care, reported 7% CP in survivors, but 36% loss to follow-up.

### Conclusions

Data from the adequately described TH studies in SSA indicate outcomes at discharge and twelve months which are similar to global norms. However, these data are limited

**Data availability statement:** All relevant data are within the manuscript and publically in the cited journals, since it is a systematic scoping literature review.

**Funding:** This study was funded in part by Department of Paediatrics and Child Health, University of Cape Town (Dr Naa A Buxton-Tetteh). No additional external funding was received for this study.

**Competing interests:** The authors have declared that no competing interests exist.

to South Africa. Interpretation of other studies was limited by loss to follow-up, variable methodology and exclusion of babies with severe HIE in some studies. There is a need for standardised definitions to facilitate interpretation in TH studies.

## Introduction

Hypoxic ischaemic encephalopathy (HIE) is an acquired syndrome of neonatal encephalopathy (NE), in term and near-term babies following acute peripartum hypoxia [1]. The annual global rate of HIE is estimated at 8.5 per 1000 births, compared to 14.9 per 1000 live births in Sub-Saharan Africa (SSA) [2,3], however, data are confounded by different case definitions [4–6]. The severity of HIE is graded as mild, moderate or severe, determined by clinical assessment which frequently includes one or more of: the Sarnat grade [7], the modified Sarnat grade [8,9], and/or the Thompson score [10,11]. An abnormal amplitude integrated electroencephalogram (aEEG) can also be used to identify babies with moderate-encephalopathy within six hours [12–14], and a suppressed aEEG at 48 hours is associated with neuro-disability and mortality [15].

In 2010, the International Liaison Committee on Resuscitation (ILCOR) recommended Therapeutic hypothermia (TH) to treat moderate-severe HIE [16]. The ILCOR recommendation was extended to resource-limited settings in 2015 [17], and limited observational studies in South Africa have since reported favourable outcomes [18,19]. However, the hypothermia for encephalopathy in low income countries (HELIX) randomised controlled trial (RCT) of TH in tertiary neonatal units in Asia, demonstrated increased mortality with TH [20]. The authors suggested suspension of TH in low- and middle-income countries (LMICs). The HELIX study findings are in contrast to two meta-analyses showing reduced mortality with TH in LMICs [21,22]. In addition, the generalisability of the HELIX cohort is limited by illness severity, extent of intrapartum hypoxia and early seizure onset before TH [23,24]. The purpose of this study was to perform a scoping review to synthesise the literature describing the use and outcomes of TH in SSA.

### Objectives

The methodology followed the Preferred Reporting Items for Systematic Reviews and Meta-Analyses extension for Scoping Reviews (PRISMA-ScR) framework and guidelines from the Joanna Briggs Institute [25–27]. The PRISMA-ScR checklist is attached separately (S1 Checklist). The specific objectives, based on the Population/Concept/Context (PCC) framework, were:

1. Identify countries and facilities in SSA (context) that have used TH (concept) for babies with HIE (population);

2. Determine if TH was standard care (concept);

3. Describe how moderate-severe HIE was diagnosed (population);

4. Describe the characteristics of babies who received TH and the criteria for TH (population);

5. Identify the number of babies from which TH data has been derived (population);

6. Describe the TH methods and management (concept);

7. Describe the complications due to TH (concept);

8. Describe the co-morbidities in babies receiving TH (concept);

9. Establish if and how neurological outcomes are reported (concept);

10. Describe the mortality in cooled babies (concept);

11. Determine if and how quality of life outcomes were reported (concept);

12. Determine if surveys of TH practices exist and describe the methods and outcomes (concept).

## Methods

### Eligibility criteria

The eligibility criteria for inclusion and exclusion criteria were based on the primary review purpose, the specific objectives, and the PCC framework and study type (Boxes 1 and 2).

**Box 1.** Eligibility criteria based on study population, concept, and context

| Category | Inclusion criteria |
|---|---|
| Population | Multiple or individual human term and near-term (Gestation ≥ 35 weeks) babies with HIE |
| Concept | Any description of the feasibility, uptake, methods and effects of TH as treatment for HIE. |
| Context | Countries and facilities in SSA |

**Box 2. Eligibility criteria based on types of evidence**

| Inclusion Criteria | Exclusion criteria |
|---|---|
| • Primary research studies (randomised controlled trials, cohort studies, cross-sectional studies, and case reports)<br>• Protocols for planned studies<br>• Letters, opinion pieces or editorials<br>• Surveys describing the use of TH | • Letters, opinion pieces or editorials which only refer to data from primary studies or surveys which have already been included<br>• Conference proceedings<br>• Dissertations<br>• Articles that are written in non-English language that have not been professionally translated into English<br>• Reviews which only refer to data from primary studies or surveys which have already been included<br>• Full text not available |

### Search strategy and selection criteria

The search strategy was developed by all authors. The following electronic databases were searched from 1 January 1996 to 31 December 2023: Medline (Pubmed), Scopus (includes Embase), Google Scholar, and Web of Science. The primary search strategy was developed for Medline via PubMed (Box 3). Keywords with synonyms and Medical Subject Headings (MeSH) were mapped using truncation and Boolean operators. Animal studies were excluded. English or translated non-English publications were included. The electronic search was supplemented by publications known to the authors and by hand searching within the full-text publications. The search terms were remapped and translated to run on each database (Box 4).

**Box 3.** Population, concept, and context (PCC) grid showing identified search terms with truncated keywords and MeSH terms for the PubMed search

| Population | Concept | Content |
|---|---|---|
| • ("infant, new-born"[MeSH Terms] OR <br> • "Baby"[Title/Abstract]) AND <br> • ("humans"[Filter]) AND | • ("hypoxia ischaemia, brain"[MeSH Terms] OR <br> • "asphyxia neonatorum"[MeSH Terms]) AND <br> • ("hypothermia, induced" [MeSH Terms] OR <br> • "cool*" [Title/Abstract] OR <br> • "hypothermia"[Title/Abstract] OR <br> • "neuroprotection"[MeSH Terms] OR <br> • "neuroprot*"[Title/Abstract]) AND | • ("Africa South of the Sahara"[MeSH Terms] OR <br> • "developing countries"[MeSH Terms] OR <br> • "LMIC"[All Fields] OR <br> • "middle income"[All Fields] OR <br> • "low income"[All Fields]) |

**Box 4.** Keywords and MeSH terms for Scopus, Google Scholar and Web of Science

| Scopus | (INDEXTERMS ("infant, newborn") OR TITLE-ABS (baby) AND NOT INDEXTERMS (animal OR animals)) AND (INDEXTERMS ("hypoxia ischemia, brain" OR "asphyxia neonatorum") AND INDEXTERMS ("hypothermia, induced" OR "neuroprotection") OR TITLE-ABS ("cool*" OR "hypothermia" OR "neuroprot*")) AND (INDEXTERMS ("Africa South of the Sahara" OR "developing countries") OR ALL ("LMIC" OR "middle income" OR "low income")) |
|---|---|
| Google Scholar | ((intitle:Infant) OR (intitle:Newborn) OR (intitle:Neonate) OR (intitle:Neonatal)) AND ((intitle:Encephalopathy) OR (intitle:Asphyxia)) AND ((intitle:Hypothermia) OR (intitle:Cooling) OR (intitle:Neuroprotection)) AND ("Africa" OR "income") |
| Web of Science | (((TS = (Infant* OR newborn* OR neonat* OR Baby OR Babies)) AND TS = (Encephalopathy OR Asphyxia OR Hypox*)) AND TS = (Hypothermia OR Cool* OR Neuroprot*)) AND TS = ("Africa South of the Sahara" OR "Africa" OR "LMIC" OR "middle income" OR "low income" OR "Saharan Africa" OR "developing countries" OR "Angola" OR "Benin" OR "Botswana" OR "Burkina Faso" OR "Burundi" OR "Cameroon" OR "Cape Verde" OR "Central African republic" OR "Chad" OR "Comoro islands" OR "Congo" OR "Brazzaville" OR "Democratic republic of Congo" OR "Cote d Ivoire" OR "Djibouti" OR "Equatorial guinea" OR "Eritrea" OR "Ethiopia" OR "Gabon" OR "Gambia" OR "Ghana" OR "Guinea" OR "Guinea Bissau" OR "Kenya" OR "Lesotho" OR "Liberia" OR "Madagascar" OR "Malawi" OR "Mali" OR "Mauritania" OR "Mauritius" OR "Mozambique" OR "Namibia" OR "Niger" OR "Nigeria" OR "Rwanda" OR "Sao tome e Principe" OR "Senegal" OR "Seychelles" OR "Sierra Leone" OR "Somalia" OR "South Africa" OR "South Sudan" OR "Sudan" OR "Swaziland" OR "Tanzania" OR "Togo" OR "Uganda" OR "Western Sahara" OR "Zaire" OR "Zambia" OR "Zimbabwe") |

## Evidence screening (selection of sources of evidence)

Endnote™20 was used to screen and de-duplicate results. Titles and then abstracts were screened to identify potentially eligible full-texts, by AH and NBT together, and independently by SP. The same reviewers screened the full-texts, to confirm their eligibility for inclusion. At each stage, the number of excluded publications were recorded and the reasons for exclusion were recorded during full text review. Disagreements were resolved by consensus or in discussion with GK.

## Data charting (extraction)

A draft data-extraction table was developed by all reviewers and piloted by reviewers NBT and SP. Charting the results was an iterative process resulting in an updated table (S1 File), which

was used for independent extraction from all publications by reviewers NBT and SP. Disagreements were resolved by consensus or in discussion with AH and GK.

### Data synthesis and presentation

The extracted data were categorised into location and study type and presented in tables aligning with the objectives using descriptive methods and narrative synthesis. Consistent with PRISMA-ScR guidelines, we did not appraise the statistical methodology or risk of bias, and we did not provide summary measures. However, the descriptions of populations and TH methods were evaluated to determine variation, validity and potential comparison with other studies.

### Ethics, registration and funding

The authors did not receive funding. Since data were publicly accessible, an ethics waiver was granted by the Human Research Ethics committee of the University of Cape Town Health Sciences Faculty (HREC/REF: 789/2023). The draft protocol was registered on the Open Science Framework, accessible at https://osf.io/r374b/?view_only=142c9000e7664f2f9cef386c603df5c9

## Results

The screening and selection process is shown in the PRISMA diagram (Fig 1). After full text review, 82 of 105 publications were excluded; 60 did not meet sufficient inclusion criteria, and 21 reviews/opinions/editorials and one dissertation referred to studies which were already included as primary publications. Thirty publications were included; 23 from database review and seven from hand-searches and prior knowledge. The publications included three surveys [34,39,48], one feasibility study [32] and 26 TH publications

### Countries, facilities and numbers of cooled babies in TH studies in SSA (Objectives 1 and 5)

The 26 publications describing TH, comprised 23 TH studies of 21 cohorts including 1420 cooled babies: 1305 in South Africa, 102 in Uganda, and 13 in Ghana (Table 1). Cooling was provided in the NICU in 86% (18/21) of cohorts. Six publications reported different aspects of three main studies, which have been reported as combined studies; Horn et al. described the cooled babies [38] within a larger HIE study of cooled and normothermic babies [11], the publications by Kali et al. in 2015 [40] and 2016 [18] reported different details from the same main study; and Robertson et al. published the protocol [44] and outcomes [45] for their pilot RCT separately. Three other publications studied different populations within a common single cohort and are reported as separate studies; Ballot et al. [30] and Sebetseba et al. [53] were included in the cohort of Simpson et al. [47].

The publications describing TH included five types of studies. Five publications described five studies of babies recruited at follow-up (survivor studies) [19,28,30,41,54]; three were small studies with selective sampling [28,41,54] and two were larger time-limited follow-up studies [19,30]. Six publications described five pilot studies of novel cooling methods in South Africa [35,36,49], Ghana [33], and Uganda [44,45]. One publication described a RCT of TH with or without morphine [50]. One publication was a case series including only two cooled babies [52]. Thirteen publications described 11 observational studies from birth of 10 cohorts of babies treated with TH (birth cohort studies) – 10 South African studies and one Ugandan study [32].

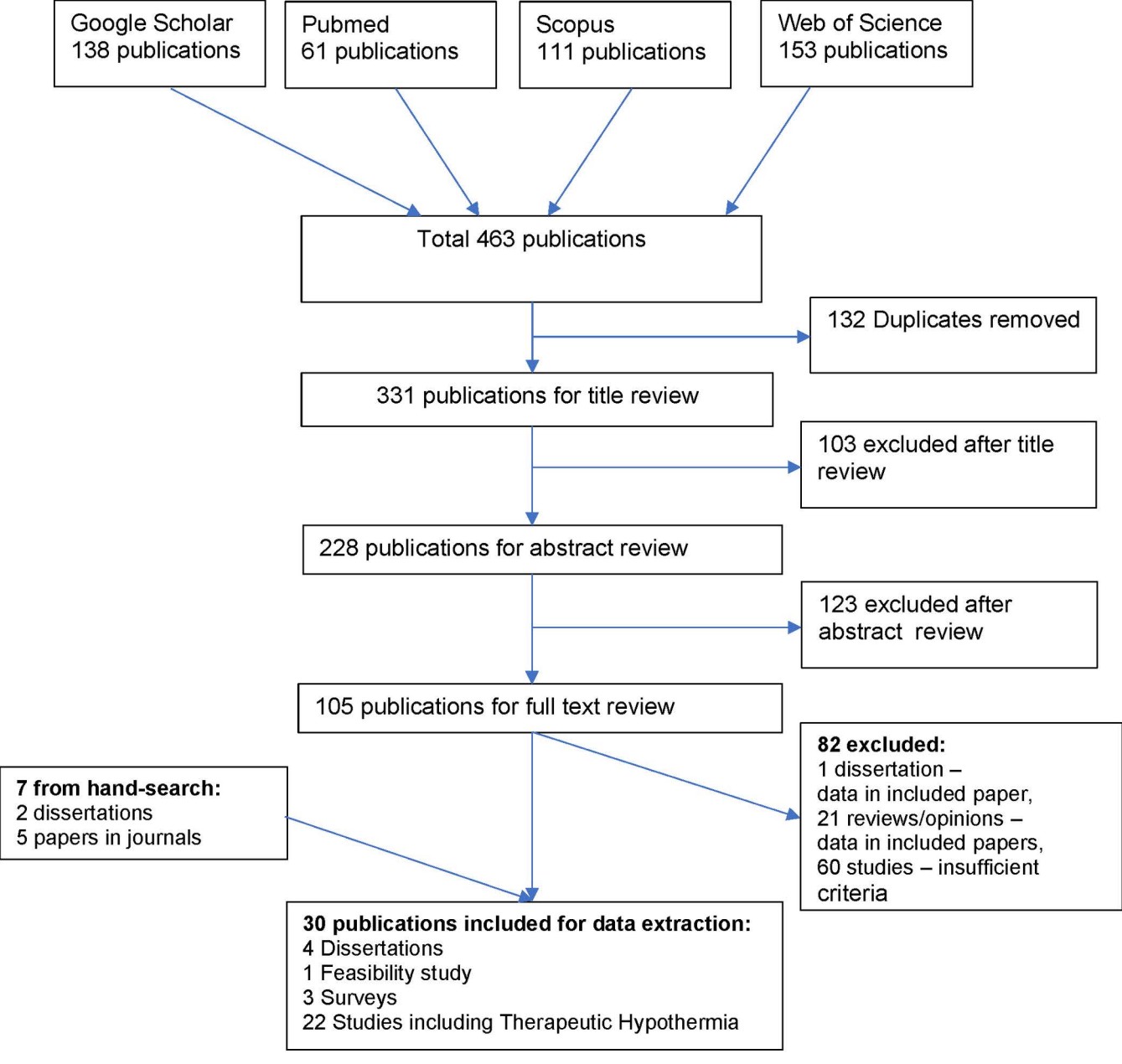

**Fig 1. PRISMA diagram showing selection of publications on TH in SSA.**

## Methods and outcomes of surveys and feasibility studies of TH practices (Objective 12)

The surveys and feasibility studies referring to TH, had diverse settings, objectives, and methods (Table 2). The opinions and TH practice of South African paediatricians were surveyed in 2012; only 37% responded, of whom 42% used TH but 75% had access or planned to use

Table 1. Characteristics of publications including cooled babies in sub-Saharan Africa.

| 1st Author, year Data year(s) | Country City Level of care | Study Design Type of Cohort | Summarised aims (sub study/overlap details) | Sample type and size |
|---|---|---|---|---|
| [a] Abrahams, 2018 [28] Data: 2010–2014 | South Africa Cape Town NICU | Retrospective case-control **Survivor study** | To study the relationship between placental pathology and neurological outcome in infants with NE and TH. | Cooled HIE: N = 17 Other HIE: N = 0 Non-HIE control: N = 30 |
| Ballot, 2020 [30] Data: 2013–2016 | South Africa Johannesburg High care | Retrospective case-control **Survivor study** | To determine developmental outcomes in survivors of HIE (all grades) compared to healthy term babies. (June 2013–Dec 2016: overlaps with Sebetseba and Simpson) | Cooled HIE: N = 49 Other HIE: N = 35 Non-HIE control: N = 64 |
| Krüger, 2019 [41] Data year unknown | South Africa Pretoria NICU | Prospective case-control **Survivor study** | To identify symptoms of oropharyngeal dysphagia in breastfeeding neonates with HIE on TH. | Cooled HIE: N = 28 Other HIE: N = 0 Non-HIE control: N = 30 |
| Mbatha, 2021[19] Data: 2013–2014 | South Africa Johannesburg High care | Retrospective descriptive **Survivor study** | To determine the neurodevelopmental outcome in babies with HIE managed with TH outside NICU. | Cooled HIE: N = 113 Other HIE: N = 0 Non-HIE/control: N = 0 |
| Stark, 2020 [54] Data: 2010–2014 | South Africa George NICU | Retrospective descriptive **Survivor study** | To describe developmental outcomes of infants with HIE who received TH at George Hospital from the ages of 3–60 months. | Cooled HIE: N = 30 Other HIE: N = 0 Non-HIE/control: N = 0 |
| Enweronu-Laryea, 2019 [33] Data: 2017 | Ghana Accra Level unclear | Prospective descriptive **Pilot study** | To assess temperature response in relation to outcome during 80 h after birth in babies with NE undergoing facilitated passive cooling as standard management. | Cooled HIE: N = 13 Other HIE: N = 0 Non-HIE/control: N = 0 |
| Horn, 2006 [49] Data: 2000–2001 | South Africa Cape Town NICU | Prospective descriptive **Pilot study** | To study the safety and efficacy of a simple, cost-effective method of selective head cooling with mild systemic hypothermia in new-born infants with HIE. | Cooled HIE: N = 4 Other HIE: N = 0 Non-HIE/control: N = 0 |
| Horn, 2009 [35] Data: 2006–2008 | South Africa Cape Town NICU | Prospective descriptive **Pilot study** | To describe an inexpensive, computer-controlled prototype fan and to report the methodology and efficacy when using the fan to cool infants with HIE to 33–34°C. | Cooled HIE: N = 10 Other HIE: N = 0 Non-HIE/control: N = 0 |
| Horn, 2010 [36] Data: 2009 | South Africa Cape Town NICU | Prospective descriptive **Pilot study** | To describe and evaluate a simple method of neuroprotective hypothermia for infants with HIE using gel-packs, a heatshield over the head and a radiant warmer to servo-control core temperature to 34°C. | Cooled HIE: N = 5 Other HIE: N = 0 Non-HIE/control: N = 0 |
| Robertson, 2008 [45] + 2011 [44] Data: 2007 | Uganda Kampala Special care | Randomised controlled **Pilot study** | To determine feasibility of consent, neurological assessment, randomisation, and TH to core temperature 33–34°C for 72 h in a low-resource setting, using water bottles. (2008 and 2011 studies are the same cohort) | Cooled HIE: N = 21 Control HIE: N = 15 Non-HIE: N = 0 |
| [a] Kali, 2021 [50] Data: 2012–2016 | South Africa Cape Town NICU | **Randomised controlled trial** | To determine if TH + morphine improves intact survival at 18 months in infants with HIE compared to TH alone. | Cooled HIE: N = 45 Other HIE: N = 0 Cooled control: N = 22/45 |
| Padayachee, 2013 [52] Data: 2006 – 2011 | South Africa Johannesburg NICU | Retrospective descriptive **Case series** | To determine the rate and grade of HIE, survival to discharge and morbidity after discharge, of neonates with perinatal asphyxia. | Cooled HIE: N = 2 Other HIE: N = 345 Non-HIE: N = 103 |
| Adams, 2022 [29] Data: 2016–2018 | South Africa Cape Town NICU | Retrospective descriptive **Birth cohort** | To determine which obstetric healthcare worker and/or system-related modifiable factors potentially contributed to adverse neurological outcome in babies with NE and TH. | Cooled HIE: N = 118 Other HIE: N = 0 Non-HIE/control: N = 0 |
| [b] Damoi, 2020 [32] Data: 2016–2019 | Uganda Kampala NICU | Retrospective descriptive **Birth cohort** | To determine short-term outcomes and factors associated with survival among new-born infants with moderate to severe HIE and TH. | Cooled HIE: N = 81 Other HIE: N = 0 Non-HIE/control: N = 0 |
| Horn, 2012 [37] Data: 2010–2011 | South Africa Cape Town NICU | Retrospective descriptive **Birth cohort** | To describe temperature data of infants with moderate-severe HIE, cooled using gel-packs, a heatshield over the head and a radiant warmer servo-controlling core temperature to 33·5°C. | Cooled HIE: N = 14 Other HIE: N = 0 Non-HIE/control: N = 0 |
| Horn, 2013a [11] + 2013b [38] Data: 2008–2009 | South Africa Cape Town NICU | Prospective descriptive **Birth cohort** | To determine the threshold Thompson score at age 3–5 h that predicted an abnormal 6-h aEEG; To determine if a Thompson score at age 3–5 h predicts a severely abnormal aEEG at 48 h or death in cooled infants. (Sub study of Horn 2013a) | Cooled HIE: N = 41 Other HIE: N = 19 Non-HIE/control: N = 0 |
| Kali, 2015 [40] +2016 [18] Data: 2008–2011 | South Africa Cape Town NICU | Retrospective descriptive **Birth cohort** | To describe the management, characteristics, clinical course, logistical challenges, and outcomes of infants treated with TH: and determine predictors of outcome at one year. (2015 and 2016 studies are the same cohort) | Cooled HIE: N = 100 Other HIE: N = 0 Non-HIE/control: N = 0 |

*(Continued)*

**Table 1.** (Continued)

| 1st Author, year<br>Data year(s) | Country<br>City<br>Level of care | Study Design<br>Type of Cohort | Summarised aims<br>*(sub study/overlap details)* | Sample type and size |
|---|---|---|---|---|
| [a]Kirabira, 2015 [51]<br>Data: 2011–2012 | South Africa<br>Cape Town<br>NICU | Retrospective descriptive<br>**Birth cohort** | To compare abnormal outcome (mortality or severely abnormal aEEG at 48 h) before discharge between inborn and out born infants with HIE who were treated with TH. | Cooled HIE: N = 57<br>Other HIE: N = 0<br>Non-HIE/control: N = 0 |
| Maphake, 2023 [42]<br>Data: 2018–2019 | South Africa<br>Pretoria<br>NICU | Retrospective descriptive<br>**Birth cohort** | To investigate if Thompson score predicts outcome at day 7 or discharge in neonates with HIE who received TH; and to investigate correlation of Thompson score with HIE risk factors. | Cooled HIE: N = 93<br>Other HIE: N = 0<br>Non-HIE/control: N = 0 |
| Nakwa, 2023 [43]<br>Data: 2015–2019 | South Africa<br>Johannesburg<br>NICU | Retrospective descriptive<br>**Birth cohort** | To determine characteristics and mortality rate at hospital discharge in neonates with moderate-to-severe HIE from a large centre in a LMIC where TH is used as standard care. | Cooled HIE: N = 394<br>Other HIE: N = 198<br>Non-HIE: N = 264 |
| Sebetseba, 2020 [53]<br>Data: 2013–2017 | South Africa<br>Johannesburg<br>High care | Retrospective descriptive<br>**Birth cohort** | To determine if neonates with NE received TH according to protocol. *(1 Jan 2013–30 Jun2017: overlaps Ballot and Simpson)* | Cooled HIE: N = 206<br>Other HIE: N = 279<br>Non-HIE/control: N = 0 |
| Simbruner, 1999 [46]<br>Data: 1997–1998 | South Africa<br>Cape Town<br>NICU | Retrospective case-control<br>**Birth cohort** | To assess the physiological effects and adverse side-effects of induced hypothermia. | Cooled HIE: N = 21<br>Control HIE: N = 15<br>Non-HIE: N = 0 |
| [a]Simpson, 2018 [47]<br>Data: 2013–2017 | South Africa<br>Johannesburg<br>High care | Retrospective descriptive<br>**Birth cohort** | To describe the characteristics and incidences of neonates diagnosed with PA, and HIE and HIE treated with TH.<br>*(1 Jan 2013–31 Jul 2017: overlaps with Ballot and Sebetseba)* | Cooled HIE: N = 213<br>Other HIE: N = 314<br>Non-HIE: N = 112 |

aEEG, amplitude integrated electro encephalography; HIE, hypoxic ischaemic encephalopathy; h, hour(s); LMIC, lower- or middle-income country; NE, neonatal encephalopathy; NICU, neonatal intensive care unit; PA, perinatal asphyxia; TH, therapeutic hypothermia.

[a]Dissertation.

[b]Preprint.

TH – whole body hypothermia was used by 93% of those who cooled, but 37% measured unvalidated target temperatures on the skin surface and/or in the nasopharynx [39]. Singla et al. surveyed global practice of TH for mild HIE in 2022, however SSA was represented by only 1/484 clinicians [48]. The global survey by Evans et al. to determine opinions on new-born care in low-income countries (LICs), including SSA, reported consensus on withholding TH for HIE [34]. The TH feasibility study in the Democratic Republic of Congo (DRC) in 2018, in hospitals without access to neonatal ventilation, investigated age at presentation and Thompson score assessment [31]. Almost half (57/134; 43%) of the babies admitted with "perinatal asphyxia" presented by age 6 hours and 74% (42/57) had moderate-severe HIE defined by a Thompson score ≥7, however all 57 babies died [31].

### Diagnosis of HIE and inclusion criteria for TH (Objective 3 and 4)

The criteria for acute peri-partum hypoxia (APPH), HIE and TH in the 23 TH studies are shown in Table 3. Alternative terminologies for APPH included, peripartum/perinatal/intrapartum/birth asphyxia/hypoxia. Intermittent positive pressure ventilation (IPPV) at age ≥10 minutes and base deficit (BD) of ≥16 mmol/l in the first 60 minutes of life were used as criteria for APPH in the majority (16/23; 70%) – a lower BD of 10 mmol/l was used in two studies (9%). The pH threshold for APPH was ≤7 in all studies which reported it (11/23; 48%). The threshold Apgar scores which indicated APPH, were 5 or 6 in most studies which reported it both 10 and 5 minutes (11/14 (79%) and 5/6 (83%) respectively). Only 26% (6/23) of studies required abnormal intrapartum events as indicators of APPH.

All studies reported clinical assessment of HIE using one or more of the following methods: Sarnat grade 44% (10/23); modified Sarnat grade 22% (5/23); presence of clinical seizures

Table 2. Surveys and feasibility studies of TH in SSA.

| First Author, Year | Joolay, 2012 [39] | Singla, 2022 [48] | Evans, 2021 [34] | Biselele, 2018 [31] |
|---|---|---|---|---|
| Countries | South Africa | Global (South Africa was the only SSA country) | South Africa, Zimbabwe, Malawi, Asia, Central America | Democratic Republic of Congo |
| Cities/ Provinces | Multiple South African cities | Multiple global cities | Multiple global cities | Kinshasa |
| Primary aim | To determine opinions and practice of South African paediatricians (including neonatologists) regarding TH and management of moderate-severe HIE. | To determine global professional opinion and practice for the use of TH for mild HIE | To determine consensus opinion of experts in newborn care about key clinical decision algorithms to assist HCWs caring for facility-based unwell newborns in LICs: HIE was one of the four algorithms | To investigate the feasibility of providing neuroprotective treatment for NE in DRC: presentation with 6 hours and use of the TS as an inclusion criterion |
| Study type | Survey | Survey | Survey | Feasibility study |
| Study method | Web-based cross-sectional survey | Web-based cross-sectional survey | Two-step modified Delphi via skype | Observational study |
| Participants and sample size or setting | 38% response rate (any response; 109/288) All/predominantly state: 45% (47/104) All/predominantly private: 55% (57/104) Neonatologists: 24% (25/104) | All surveyed 35 countries responded. 484 Neonatal clinicians in 35 countries: LMIC – 208 (43%); HIC – 276 (57%); SSA (South Africa) – 1 (0·2%) | Physician or neonatal nurse practitioner with > 10 years neonatal experience (at least 3 years in LICs). 22 invited; 14 completed round 1, ten in round 2. Nine from HIC; five from LICs. All had work experience in Africa. | 134 babies included over 9 months in 2015: GA ≥ 36 weeks with signs of perinatal asphyxia in three public hospitals with no access to ventilation or blood gas analysis. |
| Cooling sample size | < 1 treated HIE in 3 months: 41% (43/104) 1–6 in 3 months: 22% (23/104) > 6 in 3 months: 37% (38/104) | Not applicable | Not applicable | 43% (57/103) eligible for cooling and reached hospital by age 6 hours |
| APPH criteria | Not surveyed | 96% (464) used one or more of, low AS, low pH, high BD, and/or an acute sentinel event. | Risk factors for HIE: Foetal distress; Resuscitation with BVM >5 minutes or CPR >10 minutes; and/or 5-minute AS <7 | 5-minute AS ≤ 5 or the need for resuscitation at age 10 minutes |
| Moderate-severe HIE clinical criteria/ assessment method | Not surveyed | Modified Sarnat Grade: 62% Sarnat Grade: 21%, TS: 11%, Other method: 6%. | Perform TS if GA > 37 weeks and any risk factors or signs of HIE (Convulsions; Coma; Hypotonia; Absent Moro; Absent Suck; Respiratory distress). Consider other causes if GA < 37 weeks. | TS ≥ 7 in first 6 hours in 74% (76/103), or 5-minute AS ≤ 5 + signs of perinatal asphyxia (signs not specified). |
| TH criteria | Not surveyed | Perinatal insult and at least one abnormality on the Sarnat assessment. | TH not recommended | GA ≥ 36 weeks + APPH + moderate-severe HIE |
| TH uptake | 42% (43/102) use TH 33% (34/102) refer or planning to start TH | LMIC more likely to cool; p < 0·05 (25% (54/208) vs. HIC 16% (45/276)) | Not stated | Not applicable; proposal only. |
| Cooling methods | Whole Body Hypothermia: 61% (26/43) Refrigerated gel packs: 37% (16/43) Surface/nasopharynx target: 28% (12/43) Selective Cerebral Hypothermia using CoolCap: 7% (3/43) | Methods not stated. LMIC more likely to start TH after age 6 hours (42% vs. 15%; p < 0·05) LMIC more likely to stop TH early if NE resolved (LMIC 31%; HIC 11%) | Cooling, in particular, passive cooling, was not recommended. | Planned method not described. |
| Cooling opinions | TH should be standard in tertiary units: 91% (93/102) | LMIC more likely to have a protocol (51% vs. 26%; p < 0·05) | Not recommended. | Not stated |
| aEEG use | 40% (47/93) did not use EEG or aEEG | Used more often in LMIC (50% vs. HIC 30%; p < 0·01) | Not stated | Not used |
| Cranial ultrasound use | 90% (83/93) of respondents to the question used Cranial ultrasound | Not stated | Not stated | Not used |
| CT/MRI use | MRI 21% (20/93); CT scan 15% (14/93) | Not stated | Not stated | Not used |

*(Continued)*

**Table 2.** (Continued)

| First Author, Year | Joolay, 2012 [39] | Singla, 2022 [48] | Evans, 2021 [34] | Biselele, 2018 [31] |
|---|---|---|---|---|
| Developmental assessment methods | Not surveyed | Formal assessment more likely in LMIC (94% vs. HIC 73%; p < 0·05). | Not stated | Not stated |
| Mortality | Not surveyed | Not applicable | Not applicable | All those eligible for cooling died (43%) |

Abbreviations: aEEG, amplitude integrated electro encephalography; APPH, acute peripartum hypoxia; AS, Apgar score; BD, Base deficit; BVM, Bag valve mask; CPR, cardio-pulmonary resuscitation; CT, computerised tomography; GA, gestational age; HIC, high-income country; HIE, hypoxic ischaemic encephalopathy; LIC, low-income country; LMIC, low- or middle income country; MRI, magnetic resonance imaging; NE, neonatal encephalopathy; SSA, sub-Saharan Africa; TH, therapeutic hypothermia; TS, Thompson score.

in 35% (8/23) and/or Thompson score 78% (18/23). The threshold Thompson score indicating eligibility for TH was reported in 52% (12/23); four used a score of ≥10 or ≥11, six used a score of ≥7 or ≥8, and the other studies used lower scores. One study used a Thompson score of ≥8 as a criteria for TH, but paradoxically defined moderate-severe HIE as a score of ≥11. In addition to the studies using Thompson score, the clinical criteria for TH in all 23 studies included moderate-severe HIE based on one or more of the following assessments: Sarnat grade 35% (8/23); modified Sarnat grade 22% (5/23); clinical seizures 52% (12/23); Total Body Hypothermia for Neonatal Encephalopathy (TOBY) trial [12] clinical criteria 17% (4/23); and/or abnormal aEEG 22% (5/23).

The GA criteria for TH were stated in 96% (22/23); two studies included babies ≥34 weeks' GA [47,53], and the others varied from ≥36 to ≥37 weeks. The weight criteria for TH were stated in 74% (17/23) and ranged from ≥1 800 g to >2 000 g. The age at TH initiation was ≤6 hours in most studies which reported it (13/17; 76%).

## Exclusion criteria for TH in SSA studies (Objective 4)

Sixteen studies (70%) stated exclusion criteria for TH (Table 4). Four studies based criteria on the TOBY trial, and the others described criteria based on conditions which masqueraded as HIE, confounded the outcomes or were susceptible to exacerbation by TH. Two studies also excluded babies with severe HIE [43,47].

## Procedures and standards during TH (Objectives 2, 6 and 7)

The procedures, standards and complications of TH in SSA are shown in Table 5, and detailed specific cooling methods are described in Table 6. Therapeutic hypothermia was standard at the time of the study in 83% (19/23). The most common TH methods were automated whole-body cooling with a mattress (eight studies) and servo-controlled whole-body gel-pack cooling (five studies).

Other cooling methods used in individual studies included manual cooling with cold water bottles, facilitated passive cooling, servo-controlled selective head-cooling with ice-packs, automated whole-body cooling with a servo-controlled fan, manual gel-bag cooling, and cooling with commercial phase-changing material (MiraCradle Neonate Cooler, Pluss Advanced Technologies, India). There were inadequate descriptions of TH methods, machines and/or target temperatures in five studies. The time to target temperature from onset of TH was only stated in four studies and ranged from 0.5–4 hours. The duration of TH was 72 hours in most studies which stated it (16/18; 89%). The rate of rewarming ranged from ≤0.2 to ≤0.5°C/hour in the ten studies describing it. Eight studies, including four birth cohort studies [32,37,38,51], and one RCT comparing TH with and without

**Table 3. Criteria for acute peri-partum hypoxia, hypoxic ischaemic encephalopathy and therapeutic hypothermia in sub-Saharan Africa.**

| STUDIES / First author, year(s) | APPH criteria[a] | | | | | | HIE clin assessment: [b]methods + thresholds | | | | | TH criteria: [c]TS thresholds and assessment methods | | | | | | TH maturity criteria[d] | | |
|---|---|---|---|---|---|---|---|---|---|---|---|---|---|---|---|---|---|---|---|---|
| | 10-min Apgar | 5-min Apgar | Age IPPV (min) | Early pH | Early BD (mmol/l) | IP event req. | Modif Sarnat Grade | Clin Sz | Sarnat | TS | TS MS-HIE | TS | Sarnat Grade MS | Modify Sarnat Grade MS | Clin Sz | TOBY-Clin Crit | Abn. aEEG | GA (wks) | Wt (g) | age (h) |
| **Survivor studies** | | | | | | | | | | | | | | | | | | | | |
| Abrahams, 2018 [28] | ≤6 | U | ≥10 | <7 | ≥16 | N | N | N | N | Y | ≥11 | ≥8 | N | N | N | Y | N | >36 | U | ≤6 |
| Ballot, 2020 [30] | ≤4 | U | >10 | U | ≥16 | Y | N | N | Y | N | U | U | Y | N | N | N | N | >36 | >2000 | ≤6 |
| Krüger, 2019 [41] | ≤5 | U | ≥10 | ≤7 | ≥16 | N | N | Y | N | Y | U | U | N | N | Y | N | N | ≥37 | U | <6 |
| Mbatha, 2021 [19] | ≤5 | U | ≥10 | ≤7 | ≥16 | N | N | N | Y | N | U | U | Y | N | Y | Y | N | ≥36 | >2000 | <6 |
| Stark, 2020 [54] | ≤6 | U | ≥10 | U | ≥16 | N | N | Y | N | Y | ≥11 | ≥11 | N | N | Y | N | N | ≥36 | >1800 | <6 |
| **Pilot studies** | | | | | | | | | | | | | | | | | | | | |
| Enweronu-Laryea, 2019 [33] | U | ≤5 | 0 | U | U | N | Y | Y | N | Y | ≥7 | ≥7 | N | Y | Y | N | N | ≥36 | >2000 | U |
| Horn, 2006 [49] | U | ≤6 | 0 | U | ≥10 | N | N | Y | Y | Y | U | ≥2 | N | N | Y | N | N | ≥37 | U | ≤8 |
| Horn, 2009 [35] | U | ≤6 | U | U | ≥10 | N | N | Y | Y | N | U | U | Y | N | N | N | N | ≥36 | U | ≤5 |
| Horn, 2010 [36] | ≤6 | U | ≥10 | U | ≥16 | Y | N | N | Y | Y | U | U | Y | N | Y | N | Y | ≥36 | >2000 | <6 |
| Robertson, 2008 [45] + 2011 [44] | U | ≤5 | 0 | U | U | N | N | N | Y | Y | ≥5 | ≥5 | Y | N | N | N | N | ≥36 | ≥2000 | ≤3 |
| **TH analgesia RCTs** | | | | | | | | | | | | | | | | | | | | |
| Kali, 2021 [50] | ≤7 | U | ≥10 | ≤7 | ≥16 | N | N | N | N | Y | ≥7 | ≥7 | N | N | N | Y | N | ≥36 | ≥1800 | ≤6 |
| **Case series** | | | | | | | | | | | | | | | | | | | | |
| Padayachee, 2013 [52] | U | ≤5 | U | U | U | N | N | N | Y | N | U | U | Y | N | N | N | N | U | >1800 | U |
| **Birth cohort studies** | | | | | | | | | | | | | | | | | | | | |
| Adams, 2022 [29] | U | U | ≥10 | <7 | ≥16 | N | N | Y | N | Y | ≥10 | ≥10 | N | N | Y | N | N | >36 | U | U |
| Damoi, 2020 [32] | U | ≤7 | ≥10 | ≤7 | ≥16 | N | N | Y | N | Y | ≥7 | ≥7 | N | N | Y | N | N | >36 | ≥1800 | U |
| Horn, 2012 [37] | U | U | ≥10 | U | ≥16 | Y | Y | N | N | Y | U | U | N | Y | Y | N | Y | ≥36 | >2000 | <6 |
| Horn, 2013a [11] + 2013b [38] | ≤6 | U | ≥10 | ≤7 | ≥16 | Y | Y | Y | N | Y | U | U | N | Y | Y | N | Y | ≥36 | >2000 | <6 |
| Kali, 2015 [40] +2016 [18] | ≤7 | U | ≥10 | ≤7 | ≥16 | N | N | Y | N | Y | ≥11 | ≥11 | N | N | N | Y | N | ≥36 | ≥1800 | ≤8.5 |
| Kirabira, 2015 [51] | ≤6 | U | ≥10 | ≤7 | >12 | Y | Y | Y | N | N | U | U | N | Y | Y | N | Y | ≥36 | ≥1800 | ≤6 |
| Maphake, 2023 [42] | U | U | U | <7 | >16 | N | Y | Y | N | Y | ≥11 | U | N | N | Y | N | N | ≥36 | >2000 | U |
| Nakwa, 2023 [43] | ≤5 | U | ≥10 | <7 | ≥16 | N | N | N | Y | Y | ≥10 | ≥10 | N | N | N | Y | N | ≥36 | ≥1800 | <6 |
| Sebetseba, 2020 [53] | ≤5 | U | >10 | U | >16 | N | N | N | Y | Y | U | ≥8 | Y | Y | N | N | N | ≥34 | ≥2000 | ≤6 |
| Simbruner, 1999 [46] | U | U | U | U | U | Y | N | Y | N | Y | U | U | N | N | N | Y | N | ≥37 | U | U |
| Simpson, 2018 [47] | ≤5 | U | >10 | <7 | >16 | N | N | Y | Y | Y | U | ≥7 | Y | N | N | N | N | ≥34 | >2000 | <6 |

Abbreviations: Abn, abnormal; aEEG, amplitude integrated electro encephalography; APPH, acute peripartum hypoxia; AS, Apgar score; BD, Base deficit; BVM, Bag valve mask; Clin, clinical; CPR, cardio-pulmonary resuscitation; Crit, criteria; CT, computerised tomography; g, gram; GA, gestational age; h, hour; HIC, high-income country; HIE, hypoxic ischaemic encephalopathy; IP, intra-partum; LIC, low-income country; LMIC, lower- or middle-income country; min, minute; MRI, magnetic resonance imaging; MS, moderate-severe; N, no; RCT, randomised controlled trial; req, required; Sz, seizures; TH, therapeutic hypothermia; TOBY, Total Body Hypothermia for Neonatal Encephalopathy Trial; TS, Thompson score; U, unclear/unknown/not stated; wks, weeks; Y, yes.

[a]Criteria required to assign APPH.

[b]Clinical methods used to assess HIE and TS thresholds used to indicate MS HIE.

[c]Assessment criteria and TS thresholds required to qualify for TH.

[d]Maturity criteria required to qualify for TH.

**Table 4. Exclusion criteria for therapeutic hypothermia studies in sub-Saharan Africa.**

| Exclusion criteria | Author, year |
|---|---|
| Exclusion criteria not stated | [e] Adams, 2022 [29]<br>[a] Ballot, 2020 [30]<br>[e] Damoi, 2020 [32]<br>[b] Enweronu-Laryea,2019 [33]<br>[e] Maphake, 2023 [42]<br>[e] Sebetseba, 2020 [53]<br>[e] Simbruner, 1999 [46] |
| Based on TOBY trial: surgery likely within the first three days; abnormalities indicating poor outcome | [a] Abrahams, 2018 [28] |
| Based on TOBY trial: major congenital abnormalities requiring surgery; signs suggestive of chromosomal anomaly or brain dysgenesis. | [e] Kali, 2015 [40] + 2016 [18]<br>[a] Mbatha, 2021 [19] |
| Based on TOBY trial: congenital abnormalities or abnormalities requiring surgery within the first three days; intensive care not offered because of severity of encephalopathy | [c] Kali, 2021 [50] |
| Severity of HIE: grade 3 HIE not routinely treated with therapeutic hypothermia; possible causes of low Apgar scores other than perinatal asphyxia (e.g., major birth defects and chromosomal abnormalities) | [e] Simpson, 2018 [47] |
| Major congenital abnormalities; active bleeding; sepsis; $FiO_2 > 0.5$; persistent hypoglycaemia or electrolyte abnormality | [b] Horn, 2006 [49] |
| Major congenital abnormalities; active bleeding; sepsis; $FiO_2 > 0.8$; persistent electrolyte abnormality | [b] Horn, 2009 [35] |
| Major congenital abnormalities; active bleeding; sepsis; $FiO_2 > 0.8$; persistent hypoglycaemia or electrolyte abnormality | [b] Horn, 2010 [36]<br>[e] Horn, 2012 [37] |
| Infection at birth; chorioamnionitis + prolonged rupture of membranes; chromosomal syndromes; cerebral/lethal malformations; neonatal abstinence syndrome; moribund; metabolic encephalopathy; high-dose inotropes; $FiO_2 > 0.8$; high frequency oscillation ventilation. | [e] Horn, 2013a [11] + 2013b [38] |
| Severe congenital anomaly; congenital infection; persistent pulmonary hypertension; hypotension or bleeding not responding to treatment; moribund and unlikely to benefit from cooling | [e] Kirabira, 2015[51] |
| Meningitis; metabolic or congenital disorders | [a] Krüger, 2019 [41] |
| Severity of HIE: no respiratory effort > 30 minutes post resuscitation (anaesthesia effects/ medication excluded); heart rate < 100 beats per minute 20 minutes post resuscitation; major congenital abnormalities or chromosomal abnormalities involving brain | [e] Nakwa, 2023 [43] |
| Cooling programme not commenced; other causes of low Apgar scores; chromosomal abnormalities; congenital abnormalities of the central nervous system; spinal muscular atrophy; lethal conditions | [d] Padayachee, 2013 [52] |
| Apnoea; cyanosis; asystole > 10 minutes after birth; imminent death at assessment; major malformations; symptomatic infection; out born; parents live > 20 km away or do not speak or understand English/ Ugandan, or do not understand the study | [b] Robertson, 2008 [45] + 2011 [44] |
| Major congenital abnormalities likely to affect neurological outcome; unlikely to benefit from intensive care; non-hypoxic cause of encephalopathy; inadequate facilities or staff | [a] Stark, 2020 [54] |

Abbreviations: $FiO_2$, fraction of inspired oxygen; TOBY, Total Body Hypothermia for Neonatal Encephalopathy Trial.

[a]Survivor study.

[b]Pilot study.

[c]Therapeutic hypothermia analgesia randomised controlled trial.

[d]Case series.

[e]Birth cohort study

routine analgesia [50], described validated cooling methods with target core temperatures, NICU admission for all grades of HIE, and duration of cooling according to ILCOR recommendations.

Management of HIE during TH included protocols for sedation and analgesia in 44% (10/23), anti-epileptic drugs in 44% (10/23) and fluids in 22% (5/23) – only one included a feeding protocol, which was also the only study to describe the use of Empiric antibiotics [44]. Cranial ultrasound was only accessible in 35% (8/23) of studies. Magnetic resonance imaging (MRI) was only accessible in 13% (3/23) and all reported low rates of imaging ranging from

Table 5. Standards, procedures, methods, protocols and complications during TH in sub-Saharan Africa.

| STUDIES First author, year(s) | TH Standard During Study | TH Method | Target Core Temp. °C | Core Temp. Site [e] | Time to Target Temp. [a] (h) | Duration Cool (h) | Re-warm Rate °C/h | ILCOR Method + Target core site + °C + Duration [b] | Sedation/ Analgesia [c] | ASD Protocol | Fluid Protocol | CUS done | MRI done | Empiric Antibiotics During TH | Complications Due to TH |
|---|---|---|---|---|---|---|---|---|---|---|---|---|---|---|---|
| **Survivor studies** | | | | | | | | | | | | | | | |
| Abrahams, 2018 [28] | Y | Auto-mat | 33.5 | Rectal | U | 72 | ≤ 0.5 | Y | M | Y | N | U | 5/17 (27%) | U | U |
| Ballot, 2020 [30] | Y | Auto-mat | 32–34.0 | Rectal implied | U | 72 | U | N | U | N | N | Y | N | U | U |
| Krüger, 2019 [41] | Y | U | U | U | U | 72 | U | N | U | N | N | N | U | U | Y |
| Mbatha, 2021 [19] | Y | Auto-mat | 33.5 | Rectal | U | 72 | ≤ 0.5 | Y | U | N | N | Y | U | U | U |
| Stark, 2020 [54] | Y | U Selective-HC | 34.5 | Rectal | U | 72 | U | N | U | N | N | U | U | U | U |
| **Pilot studies** | | | | | | | | | | | | | | | |
| Enweronu-Laryea, 2019 [33] | Y | F-passive | 33.5 | Rectal | U | U | U | N | U | N | N | U | U | U | Y |
| Horn, 2006 [49] | N | Servo-ice | 35.3 | Rectal+Back | 0.8–4 | 48–72 | U | N | P then M | Y | Y | Y | N | U | Y |
| Horn, 2009 [35] | N | Servo-fan | 33.5 | Rectal | 0.6–2 | 72 | ≤ 0.5 | N | P then M then C | Y | Y | Y | N | U | Y |
| Horn, 2010 [36] | Y | Servo-gel | 34.0 | Rectal+Back | 0.5–1.5 | 72 | ≤ 0.2 | Y | M | Y | N | U | U | U | N |
| Robertson, 2008 [45] +2011 [44] | N | Water bottles | 33.5 | Rectal | U | 72 | ≤ 0.5 | N | P | Y | Y[d] | Y | U | Y | U |
| **TH analgesia RCTs** | | | | | | | | | | | | | | | |
| Kali, 2021 [50] | Y | Auto-mat | 33.5 | Rectal implied | U | 72 | ≤ 0.5 | Y | Routine M in 50% | N | N | Y | 25/45 (56%) | U | U |
| **Case series** | | | | | | | | | | | | | | | |
| Padayachee, 2013 [52] | Trans. | U Cerebral-C | U | U | U | U | U | N | U | N | N | U | U | U | U |
| **Birth cohort studies** | | | | | | | | | | | | | | | |
| Adams, 2022 [29] | Y | U Head-cooling | U | U | U | U | U | N | U | N | N | U | U | U | U |
| Damoi, 2020 [32] | Y | Servo-gel | 34.0 | Back Implied | U | 72 | ≤ 0.2 | Y | U | Y | N | U | U | U | U |
| Horn, 2012 [37] | Y | Servo-gel | 33.5 | Back | ≤ 3 | 72 | ≤ 0.2 | Y | M | Y | N | U | U | U | U |
| Horn, 2013b [38] | Y | Servo-gel | 33.5 | Back | U | 72 | ≤ 0.2 | Y | P then M | Y | Y | U | U | U | U |
| Kali, 2015 [40] +2016 [18] | Y | Auto-mat | 33.5 | Rectal | U | 72 | U | N | M or ASD in 91% | Y | Y | Y | 35/100 (35%) | U | N |
| Kirabira, 2015 [51] | Y | Auto-mat or Servo-gel | 33.5 | Rectal or Back | U | 72 | ≤ 0.5 | Y | U | N | N | U | U | U | U |

*(Continued)*

**Table 5.** (Continued)

| STUDIES First author, year(s) | TH Standard During Study | TH Method | Target Core Temp. °C | Core Temp. Site [e] | Time to Target Temp. [a] (h) | Duration Cool (h) | Re-warm Rate °C/h | ILCOR Method + Target core site + °C + Duration [b] | Sedation/ Analgesia [c] | ASD Protocol | Fluid Protocol | CUS done | MRI done | Empiric Antibiotics During TH | Complications Due to TH |
|---|---|---|---|---|---|---|---|---|---|---|---|---|---|---|---|
| Maphake, 2023 [42] | Y | PCM | U | U | U | 72 | U | N | U | N | N | U | U | U | U |
| Nakwa, 2023 [43] | Y | U TH-machine | U | U | U | U | U | N | U | N | N | U | U | U | U |
| Sebetseba, 2020 [53] | Y | Auto-mat | U | Rectal implied | U | U | U | N | M then P | N | N | Y | U | U | U |
| Simbruner, 1999 [46] | Y | Manual gel | U | Nasopharynx | U | 0.5–197 | U | N | Not routinely used | Y | N | U | U | U | N |
| Simpson, 2018 [47] | Y | Auto-mat | 33.5 | Rectal implied | U | 72 | U | N | U | N | N | U | N | U | N |

Abbreviations: ASD, anti-seizure drug; Auto-mat, automated whole-body cooling with mat; C, clonidine; Cerebral-C, cerebral cooling; CUS, cranial ultrasound; F-passive, facilitated passive cooling; h, hour(s); ILCOR, International Liason Committee on Resuscitation; Insul, insulated; M, morphine; Manual gel, manual gel bag cooling; MRI, magnetic resonance imaging; N, no; P, phenobarbitone; PCM, phase-changing material; RCT, randomised controlled trial; servo-fan, automated whole-body cooling with servo-controlled fan; Servo-ice, servo-controlled selective head cooling with ice packs; Temp, temperature; TH, therapeutic hypothermia; TH-machine, non-specific automated cooling machine; Trans, transitional; U, unclear/unknown/not stated; Water bottles, manual cooling with cold water bottles; Y, yes.

[a]Time from initiation of TH to reach target temp. stated as range or threshold.

[b]Method, target core temp., temp. measurement site, and duration of TH all stated and in line with ILCOR 2010 and 2015 guidelines.

[c]Description of sedation/analgesia during TH.

[d]The only study to also mention feeding protocol.

[e]"Back" refers to insulated probe between back of baby and mattress.

**Table 6. Detailed therapeutic hypothermia methods in studies in sub-Saharan Africa.**

| TH Method Type | TH Method Other name in text | TH Method | Author, year |
|---|---|---|---|
| **Automated whole-body cooling with mat** | Tecotherm TSmed200 N (Inspiration Healthcare Ltd, Leicestershire, UK) | Automated whole-body cooling and rewarming with mattress; rectal temp. monitored continuously via probe from cooling machine; target 33.5°C. | [a] Abrahams, 2018 [28]<br>[e] Simpson, 2018 [47]<br>[e] Kali, 2015 [40] + 2016 [18]<br>[c] Kali, 2021 [50]<br>[e] Kirabira, 2015 [51] |
| **Automated whole-body cooling with mat** | Servo-controlled cooling mattress. | No details of type of mattress provided by author.<br>Target rectal temp. stated as 32–34°C | [a] Ballot, 2020 [30] |
| **Automated whole-body cooling with mat** | Criticool MITRE® (Yavne, Israel) | Automated whole-body cooling and rewarming with mattress, to target rectal temp. 33.5°C. Core (rectal) temp.: monitored continuously via probe from cooling machine | [a] Mbatha, 2021 [19]<br>[e] Sebetseba, 2020 [53] |
| **Automated whole-body cooling with mat** | Blanketrol III (Cincinnati Sub-Zero [CSZ], Cincinnati, OH) | Automated whole-body cooling with mattress, to target rectal temp. 33.5°C. Core (rectal) temp.: monitored continuously via probe from cooling machine. | [c] Kali, 2021 [50] |
| **Automated cooling-non-specified** | "Cooling Machine" | Non-specific reference to automated cooling method. Target temp. not stated. | [e] Nakwa, 2023 [43] |
| **Servo-controlled whole-body gel-pack cooling** | Servo-controlled whole-body gel-pack cooling; warmer and core target 34°C | Cool gel-packs at 7-–10°C, applied to head and upper body and replaced hly. Core temp. (site not defined) servo-controlled to 34°C by overhead radiant warmer (*Servocrib, Servocare Medical Industries cc, Cape Town*) | [e] Damoi, 2020 [32] |
| **Servo-controlled whole-body gel-pack cooling** | Servo-controlled whole-body gel-pack cooling; warmer target 34°C; core target 33–34.5°C | Cool gel-packs at 7-10°C, applied to head and upper body replaced hly to keep core temp. at 33–34.5°C. measured via insulated back temp. (confirmed with rectal temp.) and servo-controlled by overhead radiant warmer (*Servocrib, Servocare Medical Industries cc, Cape Town) set to* 34°C. A heat shield covered the head. Measured mean rectal temp. 33.9 ± 0.3°C; back 33.9 ± 0.4°C | [b] Horn, 2010 [36] |
| **Servo-controlled whole-body gel-pack cooling** | Servo-controlled whole-body gel-pack cooling; warmer and core target 33.5°C | Cool gel-packs at 7–10°C, applied to head and upper body replaced hly to keep core temp. at 33.5°C. measured via insulated back or rectal temp. and servo-controlled by overhead radiant warmer (*Servocrib, Servocare Medical Industries cc, Cape Town) set to* 33.5°C. A heat shield covered the head. | [e] Horn, 2012 [37]<br>[e] Horn, 2013b [38]<br>[e] Kirabira, 2015 [51] |
| **Facilitated passive cooling** | Facilitated passive cooling | Neonates dressed in nappy, nursed uncovered without overhead heater, in ambient temp. 28.3 ± 0.7°C without fans/air-conditioning. Before study: surface temp. < 34°C, linen added. During study: Rectal (core) and axillary temp. measured every min. Target rectal temp.: 33–34°C.<br>Core < 33.5°C - apply one blanket; < 33.2°C: apply second.<br>Core < 33.0°C - place neonate in incubator. | [b] Enweronu-Laryea,2019 [33] |
| **Servo-controlled selective head cooling with ice packs** | Servo-controlled selective head cooling with ice packs | Frozen 12 × 12 cm gel-pack, covered with cloth, applied over anterior fontanelle, replaced 2–3 hly to maintain target rectal and scalp temps.<br>Back skin temp. servo-controlled to target incubator temp.<br>Target back temp.: 35–35.5°C. Target scalp temp.: 10–28°C | [b] Horn, 2006 [49] |
| **Automated whole-body cooling with a servo-controlled fan** | Automated whole-body cooling with servo-controlled fan | Rectal temp. was simultaneously servo-controlled to target measured on low-reading radiant warmer and a servo-control fan blowing cephalo-caudally, where fan speed automatically changed proportionately to rectal temp. and the radiant warmer responded to rectal temp. by automatically increasing or decreasing power according to the set target temp. A heat shield covered the head. Target Rectal temp.: 33.4–33.7°C. Achieved Rectal temp.: 3.3–34°C | [b] Horn, 2009 [35] |
| **Phase-changing material** | MiraCradle Neonate Cooler (Pluss Advanced Technologies, India) | Cooling with MiraCradle gel packs. No further details were stated by the author; however, this is a validated manual method of cooling involving regular replacement of solid phase changing material gel packs under the baby. | [e] Maphake, 2023 [42] |
| **Manual cooling with water bottles** | Manual cooling with water bottles | Manual Whole-body cooling with mattress made of three water bottles filled with tepid tap water and covered in cotton sheet. Infants wore only a nappy and were naked or wrapped in cotton sheets or blankets. Core (rectal) temp. was maintained by adding or removing sheets, blankets, or water bottles. (*Tap water temp. was stable throughout the 24 hours and around 22–25°C*) Rectal, axillary, ambient and water bottle surface temps were monitored continuously. Target rectal temp.: 33–34°C. | [b] Robertson, 2008 [45] + 2011 [44] |

*(Continued)*

**Table 6.** (Continued)

| TH Method Type | TH Method Other name in text | TH Method | Author, year |
|---|---|---|---|
| **Manual cooling with gel-bags** | Manual cooling with gel-bags | Manual cool packs at 10°C placed around the babies' heads to form a cap; cool packs replaced when nasopharyngeal temp. > 34.5°C. Abdominal skin servocontrolled or controlled manually to radiant heater target surface skin temp. 34–35°C. Target nasopharyngeal and surface skin: 34–35°C | [e] Simbruner, 1999 [46] |
| **Selective head or cerebral cooling – non-specified** | "Selective head cooling" or "Cerebral cooling" | Process/ equipment not described. Target rectal temp. 34–35°C specified by one author, not specified by the other. The only validated method of head cooling is with Cool-Cap (*Olympic Medical, Seattle, WA, USA*) which was not mentioned. | [a] Stark, 2020 [54] [d] Padayachee, 2013 [52] |

Abbreviations: hly, hourly; temp., temperature; TH, therapeutic hypothermia.

[a]Survivor study.

[b]Pilot study.

[c]TH analgesia RCT.

[d]Case series.

[e]Birth cohort study.

27–56% [18,41,50]. The MRIs done in studies by Kali et al. in 2021 and 2016 [18,50], which were matched to outcomes, were performed at widely varying times from age 7–256 and 7–145 days, with variable PPVs of 95% and 75% respectively.

Complications due to TH were reported by 17% (4/23). Cooling was stopped in a pilot study at 40 hours due to deteriorating coagulopathy in one baby [35], Kruger et al. stopped cooling in 14% at 1–48 hours, due to instability [41], and two studies reported excessive temperature variation; high incubator temperatures in the solid ice-pack study [49], and low core temperature < 33.0°C for 11 ± 18%, and > 34°C for 71 ± 22% of the time in the facilitated passive-cooling study [33].

### Admission characteristics of babies who receive TH in SSA studies (Objective 4)

Admission characteristics of babies in the 23 TH studies are shown in Table 7. Maternal characteristics were infrequently reported: intrapartum events in 39% (9/23); maternal HIV in 48% (11/23); and clinical chorioamnionitis in one study. Birth site and sex were reported by 48% (11/23) and 44% (10/23) respectively; the proportion of inborn births ranged from 33–100%, in the 11 studies with these data, but 8/11 (73%) studies included more than 79% inborns, and four studies did not include any outborn babies. The association between birth site and outcome, was reported in two of the four birth cohort studies with adequately described cooling methods in line with ILCOR recommendations [32,51]; only one study showed an independent association between birth site and outcome, where inborn babies were 74% less likely to die (Hazard Ratio: 0.26, 95% CI: 0.07, 0.94) [32].

Birth weight and gestation statistics were reported by 78% (18/23) and 52% (12/23) respectively; averages were reported variably as medians or means and ranged from 2880–3420 g and 37.7–40.3 weeks. Average 5-minute Apgar scores varied from 3–5 and were reported by 65% (14/23). Nine studies reported availability of blood gas in the first hour, ranging from 60–100%. Average (median or mean) pH was 6.96–7.24, and average BD (median or mean) was 11.5–19.9 mmol/l. Seven studies reported HIE severity pre-cooling; severe HIE or suppressed aEEG was present in 11–61% and varied with assessment methods.

**Table 7. Admission characteristics of babies treated with TH in sub-Saharan Africa.**

| STUDIES First author, year(s) | IP Events[a] | Mother HIV + [a] | Clin. Chorio[a] | Inborn[a] | Male[a] | Measured Birth Weight (g)[b] | Gestation at birth in weeks[b] | Apgar score at 5-minutes[b] | Early blood gas available[a] | Early pH[b,c] | Early BD[b,c] mmol/l | Severe HIE Precool[d] | Suppress aEEG[a,d] Precool |
|---|---|---|---|---|---|---|---|---|---|---|---|---|---|
| **Survivor studies** | | | | | | | | | | | | | |
| Abrahams, 2018 [28] | 53% (9/17) | 12% (2/17) | 41% (7/17) | U | U | 3209 ± 406 | 39.5 ± 1.5 | 4.7 ± 1.3 | 82% (14/17) | 6.96 ± 0.18 | 16.7 ± 4.2 | U | U |
| Ballot, 2020 [30] | U | U | U | U | U | U | U | U | U | U | U | U | U |
| Krüger, 2019 [41] | U | 29% (8/28) | U | U | 57% (16/28) | 3420 ± 450 | 40.3 ± 1.7 | U | U | U | U | U | U |
| Mbatha, 2021 [19] | U | U | U | U | 52% (59/113) | 3129 ± 480 | U | 5 ± 1.8 | U | U | U | U | U |
| Stark, 2020 [54] | 80% (24/30) | 13% (4/30) | U | 100% (30/30) | 50% (14/28) | 3280 (range; 2320–4360) | U | ≤ 5 in 71% (20/28) | U | U | U | U | U |
| **Pilot studies** | | | | | | | | | | | | | |
| Enweronu-Laryea, 2019 [33] | U | U | U | U | 69% (9/13) | 3138 ± 463 | 39.7 ± 1.7 | 4.9 ± 1.7 | U | U | U | Sarnat Gd 3 23% (3/13) | U |
| Horn, 2006 [49] | U | U | U | U | U | 2730–3100 | U | U | U | U | 12–16.5 | U | U |
| Horn, 2009 [35] | 100% (10/10) | U | U | U | U | 3215 (range; 2030–3925) | U | 3.0 (range; 1–6) | 100% (10/10) | U | 17.3 (range; 13–25) | Sarnat Gd 3 40% (4/10) | 50% (5/10) |
| Horn, 2010 [36] | U | U | U | U | U | 3060 (range; 2700–3640) | U | 5.0 (range; 3–6) | 60% (3/5) | U | range only; 15.9–19.5 | U | 60% (3/5) |
| Robertson, 2008 [45] + 2011 [44] | U | 14% (3/21) | U | 100% (21/21) | U | 3300 ± 500 | 38 ± 1.5 | 4.7 (no SD) | U | U | U | U | U |
| **TH analgesia RCTs** | | | | | | | | | | | | | |
| Kali, 2021 [50] | 29% (13/45) | 9% (4/45) | U | 47% (21/45) | 64% (29/45) | To: 3172 ± 466 M: 3263 ± 646 | T: 38.3 ± 1.5 M: 38.9 ± 1.5 | < 5 in 33% (15/45) | U | < 7 in 51% (23/45) | U | TS ≥ 15: 16% (7/45) | U |
| **Case series** | | | | | | | | | | | | | |
| Padayachee, 2013 [52] | U | U | U | U | U | U | U | U | U | U | U | U | U |
| **Birth cohort studies** | | | | | | | | | | | | | |
| Adams, 2022 [29] | 16% (19/118) | 16% (20/118) | U | 100% (118/118) | U | 3086 ± 576 | U | U | U | U | U | U | U |
| Damoi, 2020 [32] | 66% (53/81) | U | U | 79% (64/81) | 67% (54/81) | U | ≥ 40: 38% (30/81) | < 5 in 29% (23/81) | U | U | U | TS ≥ 15: 11% (9/81) | U |
| Horn, 2012 [37] | U | U | U | U | U | 3078 (range; 2140–3570) | 38.0 (range; 36–42) | U | U | U | U | U | U |
| Horn, 2013b [38] | 93% (38/41) | 66% (27/38) | U | 90% (37/41) | 56% (23/41) | 3220 ± 407 | U | 4.0 (IQR; 3–5) | 73% (30/41) | 7.02 ± 0.16 | 17.6 (range; 5.9–22.6) | MS Gd 3: 32% (13/41) | 61% (25/41) |
| Kali, 2015 [40] + 2016 [18] | 13% (13/98) | 21% (19/89) | U | 33% (33/100) | 59% (59/100) | 3060 (range; 1960–5190) | 39.0 (range; 35–43) | 4.5 (range; 0–8) | 81% (81/100) | 7.03 ± 0.15 | U | U | U |
| Kirabira, 2015 [51] | 58% (33/57) | 29% (16/56) | U | 40% (23/57) | U | U | U | U | 93% (53/57) | U | IB: 16.7 ± 8.4 OB: 16.7 ± 5.9 | U | 40% (22/55) |

*(Continued)*

Table 7. (Continued)

| STUDIES First author, year(s) | IP Events[a] | Mother HIV +[a] | Clin. Chorio[a] | Inborn[a] | Male[a] | Measured Birth Weight (g)[b] | Gestation at birth in weeks[b] | Apgar score at 5-minutes[b] | Early blood gas available[a] | Early pH[b,c] | Early BD[b,c] mmol/l | Severe HIE Precool[d] | Suppress aEEG[a,d] Precool |
|---|---|---|---|---|---|---|---|---|---|---|---|---|---|
| Maphake, 2023 [42] | U | 19% (18/93) | U | 100% (93/93) | U | U | U | U | 100% (93/93) | U | U | U | U |
| Nakwa, 2023 [43] | U | U | U | U | U | 3058±508 | 38.1±2.5 | 5 (IQR; 4–6) | 100% (394/394) | 7.05 (IQR; 6.90–7.16) | 19.9 (IOR; 16–24) | U | U |
| Sebetseba, 2020 [53] | U | U | U | 79% (163/206) | 62% (127/206) | 2880±660 | 38.3±2.3 | 4.8 (range; 0–8) | | 7.1±0.2 | 17.5±6.3 | U | U |
| Simbruner, 1999 [46] | U | U | U | U | U | 3087±488 | 39.4±1.8 | 5.2±2.3 | 76% (16/21) | 7.24±0.18 | 11.5±7.2 | U | U |
| Simpson, 2018 [47] | U | 24% (50/213) | U | 81% (173/213) | 62% (132/213) | 3071±478 | 38.7±1.9 | U | U | U | U | U | U |

Abbreviations: aEEG, amplitude integrated electro encephalography; AS, Apgar score; BD, base deficit; Chorio, chorioamnionitis; Clin, clinical; Gd, grade; HIE, hypoxic ischaemic encephalopathy; HIV, human immunodeficiency virus; IB, inborn; IP, intrapartum; IQR, interquartile range; MS, modified Sarnat; OB, out born; RCT, randomised controlled trial; SD, standard deviation; TH, therapeutic hypothermia; U, unclear/unknown/not stated in cooled babies.

[a]Proportion reported.

[b]Statistics as reported in publications as mean ± SD, median (IQR) or (range).

[c]'Early' refers to first 60 minutes or cord.

[d]Worst grade/severity before cooling.

**Table 8. Co-morbidities of babies treated with TH in sub-Saharan Africa.**

| STUDIES First author, year(s) | IIPPV | NICU available for severe HIE | Inotropes | Acquired infection | NEC | Hypogly-caemia | Hyper-glycaemia | Bleeding Treated | PPHN |
|---|---|---|---|---|---|---|---|---|---|
| **Survivor studies** | | | | | | | | | |
| Abrahams, 2018 [28] | 59% (10/17) | Y | 6% (1/17) | U | U | U | U | U | U |
| Ballot, 2020 [30] | IIPV not available | N | Inotropes not available | U | U | U | U | U | U |
| Krüger, 2019 [41] | U | U | U | U | U | U | U | U | U |
| Mbatha, 2021[19] | 10% (11/113) | N | U | U | U | U | U | U | U |
| Stark, 2020 [54] | U | U | U | U | U | U | U | U | U |
| **Pilot studies** | | | | | | | | | |
| Enweronu-Laryea, 2019 [33] | 0% | U | U | 0% | U | U | U | U | U |
| Horn, 2006 [49] | 0% | Y | 0% | 0% | U | 25% (1/4) | 25% (1/4) | U | U |
| Horn, 2009 [35] | 50% (5/10) | Y | 50% (5/10) | 10% (1/10) | U | 20% (2/10) | U | 10% (1/10) | U |
| Horn, 2010 [36] | 40% | Y | 40% | U | U | 20% | U | U | 20% |
| Robertson, 2008 [45] + 2011 [44] | IIPV not available | N | IIPV not available | U | U | U | U | U | U |
| **TH analgesia RCTs** | | | | | | | | | |
| Kali, 2021 [50] | 60% (27/45) | Y | 22% (10/45) | U | 4% (2/45) | 38% (17/45) | 24% (11/45) | U | U |
| **Case series** | | | | | | | | | |
| Padayachee, 2013 [52] | U | N | U | U | U | U | U | U | U |
| **Birth cohort studies** | | | | | | | | | |
| Adams, 2022 [29] | U | Y | U | U | U | U | U | U | U |
| Damoi, 2020 [32] | 20% (16/81) | Y | U | U | U | U | U | U | U |
| Horn, 2012 [37] | U | Y | U | U | U | U | U | U | U |
| Horn, 2013b [38] | 61% (25/41) | Y | 34% (14/41) | 2% (1/41) | U | U | U | U | 7% (3/41) |
| Kali, 2015 [40] + 2016 [18] | 34% (34/100) | Y | 17% (13/77) | U | 1% (1/99) | 4% (3/77) | 10% (8/77) | 8% (5/77) | U |
| Kirabira, 2015 [51] | 35% (20/57) | Y | 16% (9/57) | 4% (2/57) | 2% (1/57) | 16% (9/57) | U | U | 9% (5/57) |
| Maphake, 2023 [42] | U | Y | 38% (35/93) | U | U | U | U | U | U |
| Nakwa, 2023 [43] | U | N | U | U | U | U | U | U | U |
| Sebetseba, 2020 [53] | IIPV not available | N | IIPV not available | U | U | U | U | U | U |
| Simbruner, 1999 [46] | 62% (13/21) | Y | U | U | U | U | U | U | U |
| Simpson, 2018 [47] | IIPV not available | N | IIPV not available | 6% (12/213) | U | U | U | U | 2% (4/213) |

Abbreviations: IIPPV, intubated intermittent positive pressure ventilation; NEC, necrotising enterocolitis; N, no; PPHN, persistent pulmonary hypertension; RCT, randomised controlled trial; TH, therapeutic hypothermia; U, unclear/unknown/not stated in cooled babies; Y, yes.

## Co-morbidities of babies who receive TH in SSA studies (Objective 8)

The co-morbidities of babies in the 23 TH studies are shown in Table 8. Intubated intermittent positive pressure ventilation (IIPPV) was not available in four studies and was unclear in seven studies. Most studies with IIPPV (8/10; 80%), ventilated ≤ 50% of babies. Treatment with inotropes was reported in 39% (9/23). The rate of acquired infection and/or NEC ranged from 0–10% in the six studies reporting it. Treated bleeding was only reported by two studies and occurred in ≤ 10%. Pulmonary hypertension and glucose abnormalities were variable but infrequently reported, occurring in 2–20% in four studies, and 4–25% in six studies, respectively.

**Table 9. Hospital stay and neurological and mortality outcomes of babies treated with therapeutic hypothermia in sub-Saharan Africa.**

| STUDIES First author, year | Severe HIE[a] (Gd 3) | Duration Hospital Stay (days)[b] | Time to full cup/breast feed (days)[b] | Died Before DC or 12-month follow-up[c] | Survived to DC + non-suppressed 48-h aEEG | Assessment method at 12-months | Other or additional assessment methods used | Survived + normal or mildly abnormal at 12 months | Died or CP at 12 months | CP in survivors at 12 months | Lost to follow up 12 month[d] |
|---|---|---|---|---|---|---|---|---|---|---|---|
| **Survivor studies** | | | | | | | | | | | |
| Abrahams, 2018 [28] | 18% (3/17) | U | U | 0 | U | BSID-III | U | 44% (7/16) | 56% (9/16) | 44% (7/16) | 6% (1/17) |
| Ballot, 2020 [30] | U | U | U | U | U | BSID-III (14 months) | U | U | U | 10% (5/49) | U |
| Krüger, 2019 [41] | 7% (2/28) | U | U | U | U | U | U | U | U | U | U |
| Mbatha, 2021 [19] | 10% (12/113) | U | U | U | U | GMDS | GMDS at 18-24 months | U | U | 7% (5/72) | 36% (41/113) |
| Stark, 2020 [54] | 29% (8/28) | U | U | 3% (1/30) | U | START | ECDC at 4 and 5 years GMFCS-E&R from 3 months to 5 years | 74% (20/27) | 33% (9/27) | 31% (8/26) | 10% (3/30) |
| **Pilot studies** | | | | | | | | | | | |
| Enweronu-Laryea, 2019 [33] | 23% (3/13) | 10±4.6 | U | 23% (3/13) | U | U | U | U | U | U | U |
| Horn, 2006 [49] | 0 | U | U | 0 | U | U | U | U | U | U | U |
| Horn, 2009 [35] | 40% (4/10) | U | U | 10% (1/10) | U | U | U | U | U | U | U |
| Horn, 2010 [36] | 40% (2/5) | 60% by D10 (3/5) | U | 0 | U | U | U | U | U | U | U |
| Robertson, 2008 [45] + 2011 [44] | 29% (6/21) | 8±2.3 | U | 33% (7/21) | U | U | Modified optimality score at 7 and 17 days | U | U | U | U |
| **TH analgesia RCTs** | | | | | | | | | | | |
| Kali, 2021 [50] | 16% (7/45) | U | U | 13% (6/45) | U | BSID-III | BSID III at 18 months; Amiel-Tison at 3 months; OAE, BAER | 71% (27/38) | 32% (12/38) | 16% (5/32) | 16% (7/45) |
| **Case series** | | | | | | | | | | | |
| Padayachee, 2013 [52] | U | U | U | 0 | U | U | U | U | U | U | 100% (2/2) |
| **Birth cohort studies** | | | | | | | | | | | |
| Adams, 2022 [29] | U | 13 (mean, no SD) (range: 3-103) | U | 11% (13/118) | U | U | U | U | U | U | 10% (11/113) |
| Damoi, 2020 [32] | 10% (8/81) | 7 (IQR) (4-34) | 6 (IQR) (5-8) | 16% (13/81) | U | U | U | U | U | U | U |
| Horn, 2012 [37] | U | U | U | 21% (3/14) | U | U | U | U | U | U | U |
| Horn, 2013b [38] | 51% (21/41) | U | U | 20% (8/41) | 71% (29/41) | U | U | U | U | U | U |
| Kali, 2015 [40] + 2016 [18] | 23% (22/97) | 9 (range) (5-46) | U | 17% (17/100) | U | BSID-III | BSID III and Amiel-Tison at 3 months | 61% (41/67) | 39% (26/67) | 18% (9/50) | 33% (33/100) |

(Continued)

**Table 9.** (Continued)

| STUDIES First author, year | Severe HIE[a] (Gd 3) | Duration Hospital Stay (days)[b] | Time to full cup/ breast feed (days)[b] | Died Before DC or 12-month follow-up[c] | Survived to DC + non-suppressed 48-h aEEG | Assessment method at 12-months | Other or additional assessment methods used | Survived + normal or mildly abnormal at 12 months | Died or CP at 12 months | CP in survivors at 12 months | Lost to follow up 12 month[d] |
|---|---|---|---|---|---|---|---|---|---|---|---|
| Kirabira, 2015 [51] | 39% (22/57) | U | U | 32% (18/57) | U | U | U | U | U | U | U |
| Maphake, 2023 [42] | 14% (13/93) | U | U | 2% (2/93) | U | U | U | U | U | U | U |
| Nakwa, 2023 [43] | 10% (40/394) | U | U | 17% (67/394) | U | U | U | U | U | U | U |
| Sebetseba, 2020 [53] | U | 8.1 ± 7.8 | U | U | U | U | U | U | U | U | U |
| Simbruner, 1999 [46] | U | U | U | 19% (4/21) | U | U | U | U | U | U | U |
| Simpson, 2018 [47] | 15% (32/213) | 7 (IQR) (0–89) | U | 12% (26/213) | U | U | U | U | U | U | U |

Abbreviations: aEEG, amplitude integrated electro encephalography; BAER, brainstem auditory evoked responses; BSID-III, Bayley scales of infant and toddler development, 3rd edition; CP, cerebral palsy; DC, discharge; ECDC, Early Childhood Developmental Criteria; Gd, grade; GMDS, Griffiths Mental Development Scales; GMFCS-E&R, Gross Motor Function Classification System-Expanded and Revised; h, hour; HIE, hypoxic ischaemic encephalopathy; IQR, interquartile range; OAE, oto-acoustic emissions; RCT, randomised controlled trial; SD, standard deviation; TH, therapeutic hypothermia; U, unclear/unknown/not stated for cooled babies.

[a]Proportion with grade 3 HIE as the highest reported grade.

[b]statistics as reported in publications as mean±SD, median (IQR) or (range).

[c]Proportion who died at discharge or by 12-months, which ever latest.

[d]Proportion of infants, thought to be alive, in whom developmental outcomes were not available.

## Hospital stays, neurological outcomes, quality of life and mortality (Objectives 9, 10 and 11)

The Hospital stays and outcomes of babies treated with TH in SSA are shown in Table 9. Quality of life outcomes were not reported. The rate of severe HIE ranged from 0–51%, and was reported in 70% (16/23) of studies. Average hospital stays, reported by eight studies, were 7–13 days. One study reported time to cup or breast feeding [32], and one study reported hearing assessments [50]. The most frequent of the six neurological assessment methods after discharge, was the Bayley scales of infant and toddler development, third edition (BSID-III), at 12 months and outcomes of the methods used prior to 12 months were not reported.

Death before discharge or 12-month follow-up ranged from 0–33%. The four birth cohort studies and the RCT of analgesia during TH that were aligned with ILCOR recommendations [32,37,38,50,51], reported mortalities of 13–32%. Two birth cohort studies reported survival with normal/mildly abnormal outcome; one study reported 71% survival with non-suppressed aEEG at 48 hours [38], and the other reported 61% survival with normal/mildly abnormal outcome at 12 months and 18% cerebral palsy (CP) in survivors, but 33% loss to follow-up and 22% were cooled between 6 and 8.5 hours [18]. However, the RCT of analgesia during TH, which recruited babies within 6 hours of birth, reported 71% overall survival with normal/mildly abnormal outcome at 12 months for both groups, and 16% cerebral palsy (CP) in survivors, with only 16% loss to follow-up [50]. The birth cohort studies that restricted TH or IIPV to babies ≥ 36 weeks' GA without severe HIE [43], or restricted TH to inborn babies [42], had low mortality rates (2–17%). Similarly, the survivor study of outcomes following TH without NICU, reported only 7% CP in survivors at 12 months, but moderate-severe disability *or* CP occurred in 13% (9/72) and loss to follow up was 36% – this was the only survivor study with non-selective sampling and cooling methods aligned to ILCOR [17].

Two studies reported outcomes at 18 months; Kali et al. [50] showed similar outcomes to their 12-month assessment (68% survival with absent or mild disability) with 16% loss to follow-up, however Mbatha et al. [19] showed increased loss to follow-up (48%) and increased moderate/severe disability or CP, which was reported in 37% (22/59).

## Discussion

This scoping review identified 30 records, including three surveys, one feasibility study and 26 publications of 23 TH studies, cooling 1420 babies. Only three studies were outside South Africa [32,33,44,45], two of which were pilot studies [33,44,45]. Therapeutic hypothermia was standard care in most South African studies and one Ugandan study [32]. There were multiple designs, overlapping cohorts, and limited follow-up data. The criteria for APPH, moderate-severe HIE, and TH were largely similar to ILCOR recommendations [17], However TH methodology and morbidity reporting was highly variable and often inadequate. The birth cohort studies and the RCT of analgesia during TH, with adequately described aspects of TH methodology in line with ILCOR recommendations [17], had mortality and intact survival rates comparable with global norms [32,37,38,50,51]. However, loss to follow-up was high in most studies and mortality rates were higher than in the birth cohort studies where TH or IIPPV was restricted to near-term or term babies without severe HIE [43], or to inborn babies only [42]. Similarly, the CP rate was low in the survivor study where babies with severe HIE had not been admitted to NICU [19].

The South African survey of TH from 2012 showed substantial non-standard temperature monitoring [39], similar to more recent surveys in India [55], and Brazil [56]. The global survey of TH established consensus against TH in LICs [34], in keeping with the findings of the feasibility study in DRC, showing high HIE rates, no access to neonatal intensive care, high mortality and substantial infrastructure challenges [31].

### Diagnosis of HIE and criteria for TH

Consistency in diagnostic criteria facilitates robust comparisons and conclusions, however there was wide diagnostic variation in this review. Intrapartum abnormalities were infrequently required as indicators of APPH, in keeping with acknowledged challenges in documenting such events [5]. The predominant indicators of APPH were resuscitation requirements based on five- or ten-minute Apgar scores, acid-base abnormalities in the first 60 minutes and IPPV for at least 10 minutes. This approach is supported by population studies of babies with HIE; one study showed strong correlation between a 5-minute Apgar score < 7 and acidosis [5], and another showed advanced neonatal resuscitation as the only independent early indicator of death [57]. The combined use of aEEG with any moderate signs of the modified Sarnat grade is most predictive for abnormal outcome [8,9], but both were infrequently used. The assessment most often used to diagnose HIE, was the Thompson Score, followed by Sarnat and modified Sarnat grades. However, the threshold Thompson score indicating moderate-severe HIE and/or a criterion for TH varied substantially. Several SSA studies used the Thompson thresholds from the study of normothermic infants where the score was assigned after days, not hours [10]. However, a Thompson score ≥ 7 at age 1–6 hours is the most predictive threshold for abnormal aEEG [11], and abnormal MRI [58], while a score of ≥ 8 misses 19% of babies with abnormal aEEG [59].

The GA, birth weights and ages at TH initiation were often in line with ILCOR[17] recommendations; however, two studies commenced TH after 6 hours in a minority babies [18,40,49], and two studies included babies at ≥ 34 weeks' GA [47,53]. There is no benefit of TH if commenced after 6 hours [60], and no benefit for preterm babies < 36 weeks' GA [61]. Exclusion criteria for TH were frequently not stated and were variable but were generally similar to those in the global RCTs showing benefit [16,62]. However, some studies did not cool or did not ventilate babies with severe HIE [19,30,43,47,52,53], and some studies only included inborn babies [29,42,44,45,54]. These approaches are supported by improved outcomes associated with early initiation of TH [63], and the increased benefit of TH with moderate HIE compared to severe HIE [62,64].

### Procedures, management, and morbidity during TH

The methodology of TH, admission characteristics and co-morbidities of cooled babies were often poorly described, with insufficient detail to determine validity of the TH method. Although there was less ventilation, inotrope use, bleeding and infection in the adequately described birth cohort studies and the RCT of analgesia during TH [18,32,37,38,40,50,51] compared to the HELIX study [20], the low reporting rate limits interpretation.

### Outcomes in cooled infants

Only three studies reported MRI use and the proportions of scanned babies were too small to draw meaningful conclusions related to outcomes. The CP rate in the largest survivor study of cooled babies in South Africa, who were cooled in line with ILCOR guidelines, but not in NICU[19], was substantially lower than rates in reported in global meta-analyses [22,62], but loss to follow-up was high. The RCT of analgesia with TH and the birth cohort studies with adequate recruitment and methodology [32,37,38,50,51] had lower mortality rates than reported in the HELIX trial [20]. Similarly, the survival with normal/mildly abnormal outcome of 71% based on 48-hour aEEG [38], and 71% based on BSIDIII at 12 months with only 16% loss to follow up [18,40], are both higher than the survival without neuro-disability in the HELIX trial and comparable to global meta-analyses [20,22]. The lower mortality in birth cohort studies which restricted TH, IIPV, or intensive care to babies ≥ 36 weeks' GA without

severe HIE [43], or restricted inclusion to inborn babies only [42], are in keeping with previous reports of increased benefit in those settings [62–64].

## Potential control groups

Seven of the studies which were included in this review also described outcomes in cohorts of non-cooled babies with moderate-severe HIE. However these cohorts should not be used as controls to inform the efficacy of TH when applied within ILCOR guidelines due to the following reasons: In one study, they were managed in a setting without high care and ventilation facilities [44]; in three studies the cohorts included preterm babies or did not specify gestational age [47,52,53]; in one study babies with HIE were not ventilated [30], in another study the criteria indicating moderate-severe HIE were not adequately described [46]; and in the final study, the majority of the non-cooled cohort consisted of babies who were not offered TH because they were deemed to be moribund (54%), too unstable for TH (10%) or too mild for TH (9%) [43]. Previous historical data of non-cooled babies in centres, prior to provide TH in SSA and in centres that do provide TH in SSA, were not the subject of this review, but may not offer appropriate control groups since the management and definition of HIE varies considerably with time and between centres [65].

## Strengths and limitations

This review has several limitations which limit generalisability and comparison of findings. First, the studies had multiple different designs and methodology, including several pilot studies and studies with small convenience sample sizes. Second, there were no large RCTs comparing TH to normothermia, and the larger studies were observational with frequently variable and inadequately described methodology. Finally, the review excluded conference proceedings. However, there are several strengths. This is the first scoping review of TH in SSA. The methodology was in accordance with PRISMA-ScR [26]. Multiple objectives provided a framework for a comprehensive review and four data bases were searched, including the grey literature in Google Scholar.

## Conclusion

This scoping review aimed to synthesise the literature describing TH in SSA, to inform opinions on the appropriateness of TH in SSA. We identified a substantial body of literature describing TH in SSA, but most were from South Africa. The surveys and feasibility studies had limited representation and applicability. The highly variable and often inadequate reporting of TH methods compromised generalisability and comparison of findings in most of the SSA studies, and demonstrates the need for standardised criteria for TH, more detailed reporting and a consensus minimum data set for TH studies in SSA. However, the two studies including babies from birth with adequately described and validated cooling methods, and non-selective sequential recruitment [38,50], and one survivor study of clinic attenders [19] showed outcomes comparable to global norms. Although the high loss to follow-up in some studies limits interpretation of outcomes, the low rate of mortality and abnormal 48-hour aEEG by discharge [38], suggests that these outcomes are likely to be representative. Notably, the study with the lowest CP rate did not admit babies with severe HIE to NICU [19]. Future studies should prioritise the identification of babies who are least likely to benefit from cooling.

## Supporting information

**S1 Checklist. PRISMA checklist.**
(PDF)

**S1 File. Final data table.**
(PDF)

## Acknowledgments

We acknowledge the support from the academic administrative staff in the Department of Paediatrics and Child Health and the Division of Neonatology, University of Cape Town.

## Author contributions

**Conceptualization:** Alan Richard Horn.

**Data curation:** Naa Akyere Buxton-Tetteh, Shakti Pillay, Alan Richard Horn.

**Formal analysis:** Naa Akyere Buxton-Tetteh, Shakti Pillay, Alan Richard Horn.

**Investigation:** Naa Akyere Buxton-Tetteh, Shakti Pillay, Gugulabatembunamahlubi Tenjiwe Jabulile Kali, Alan Richard Horn.

**Methodology:** Naa Akyere Buxton-Tetteh, Shakti Pillay, Gugulabatembunamahlubi Tenjiwe Jabulile Kali, Alan Richard Horn.

**Project administration:** Alan Richard Horn.

**Supervision:** Shakti Pillay, Alan Richard Horn.

**Validation:** Naa Akyere Buxton-Tetteh, Gugulabatembunamahlubi Tenjiwe Jabulile Kali, Alan Richard Horn.

**Writing – original draft:** Naa Akyere Buxton-Tetteh.

**Writing – review & editing:** Naa Akyere Buxton-Tetteh, Shakti Pillay, Gugulabatembunamahlubi Tenjiwe Jabulile Kali, Alan Richard Horn.

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
