## [Decision Letter · Decision Letter 0]

25 Oct 2024

PONE-D-24-28470Therapeutic hypothermia for neonatal hypoxic ischaemic encephalopathy in Sub-Saharan Africa: a scoping reviewPLOS ONE

Dear Dr. Horn,

Thank you for submitting your manuscript to PLOS ONE. After careful consideration, we feel that it has merit but does not fully meet PLOS ONE’s publication criteria as it currently stands. Therefore, we invite you to submit a revised version of the manuscript that addresses the points raised during the review process.

**ACADEMIC EDITOR:**Please follow comments from the first reviewer and answer and amend you article accordingly. Kind regards

We look forward to receiving your revised manuscript.

Kind regards,

Stefan Grosek, Ph.D., M.D.,

Academic Editor

PLOS ONE

Journal Requirements: When submitting your revision, we need you to address these additional requirements. 1. Please ensure that your manuscript meets PLOS ONE's style requirements, including those for file naming. The PLOS ONE style templates can be found at https://journals.plos.org/plosone/s/file?id=wjVg/PLOSOne_formatting_sample_main_body.pdf and https://journals.plos.org/plosone/s/file?id=ba62/PLOSOne_formatting_sample_title_authors_affiliations.pdf 2. Please include captions for your Supporting Information files at the end of your manuscript, and update any in-text citations to match accordingly. Please see our Supporting Information guidelines for more information: http://journals.plos.org/plosone/s/supporting-information. 3. Please review your reference list to ensure that it is complete and correct. If you have cited papers that have been retracted, please include the rationale for doing so in the manuscript text, or remove these references and replace them with relevant current references. Any changes to the reference list should be mentioned in the rebuttal letter that accompanies your revised manuscript. If you need to cite a retracted article, indicate the article’s retracted status in the References list and also include a citation and full reference for the retraction notice.

**Additional Editor Comments:**

Dear Authors

Interseting study which need some amendments presented by one of the reviewers.

Kind regards

Reviewers' comments:

Reviewer's Responses to Questions

**Comments to the Author**

1. Is the manuscript technically sound, and do the data support the conclusions?

Reviewer #1: Yes

Reviewer #2: Yes

2. Has the statistical analysis been performed appropriately and rigorously? 

Reviewer #1: N/A

Reviewer #2: Yes

3. Have the authors made all data underlying the findings in their manuscript fully available?

Reviewer #1: Yes

Reviewer #2: Yes

4. Is the manuscript presented in an intelligible fashion and written in standard English?

Reviewer #1: Yes

Reviewer #2: Yes

5. Review Comments to the Author

Reviewer #1: The authors in a manuscript titled "Therapeutic Hypothermia for Neonatal Hypoxic Ischemic Encephalopathy in Sub-Saharan Africa" presented the first review of the extent of TH in SSA, which aimed to identify countries and institutions in SSA that used TH through a systematic literature review, to describe whether TH is the standard of care, characteristics of newborns receiving TV, methods, complications, and outcomes. The methodology was in accordance with the PRISMA-ScR standard. Several targets provided a framework for a comprehensive review, and four databases were searched, including the grey literature in Google Scholar.

Thank you for a comprehensive overview of the state of the SSA, conclusions and directions forward. The fact is that the data are poorly comparable, there has been inconsistency in the presentation of studies, there is an overlap of studies, which all adds up to the need for standardized criteria for TH and minimum criteria for the analysis of outcomes. It is very interesting that this analysis includes N=1420 newborns treated with TH in SSA. The authors themselves point to the overlap of the research, which may also have an impact on the final conclusions. However, the three studies with adequately described and validated cooling methods, non-selective sequential recruitment, and neurological outcomes showed outcomes comparable to global norms.

My comments and questions:

1. Nowhere in the analysis is there any mention of the outcome based on MRI of the brain; even though there is aforementioned MRI done in three studies, we don't have their analysis of the extent of the brain injury?

2. In addition, studies have shown that clinical neurological examination at the end of the first month of life has predictive value regarding the outcome of newborns treated with TH at HIE. Do you have this data from your analysis? Please comment and add as this information can be very comparable to the extent that a neurological clinical analysis is performed according to a standardized methodology and a good/credible basis for comparison.

3. Do you have any data, and can you make a difference in terms of the impact of TH on reductions in death or disability, either among inborn neonates or outborn neonates?

4. Are there data on the outcome of children with moderate/severe hypoxia who are not treated with TH in SSA and can these be used as a control group?

5. Despite the fact that the results on mortality and intact survival rates are comparable with global norms. However, loss to follow-up was high and mortality rates were higher than in the birth cohort studies where TH or IIPPV was restricted to near-term or term babies without severe HIE, or to inborn babies only. Similarly, the CP rate was low in the survivor study where babies with severe HIE had not been admitted to NICU. Despite all of the above, can you conclude that the results are comparable to the global norm?

6. Do you think that the effectiveness of TH treatment would be comparable to the results from the HELIX-LIC study if this analysis also included newborns with severe HIE who would need intensive treatment? Can you please comment and add in the manuscript?

Reviewer #2: Great Job, wonderful comprehensive Data analysis

The Authors are to be congratulated on this work

Data analysis valuable Demonstration efficacy of Therapeutic Hypothermia in low-Ressource settings

No changes required

6. PLOS authors have the option to publish the peer review history of their article (what does this mean? ). If published, this will include your full peer review and any attached files.

**Do you want your identity to be public for this peer review?** For information about this choice, including consent withdrawal, please see our Privacy Policy .

Reviewer #1: No

Reviewer #2: **Yes: ** Prof Dr Sascha Meyer

---

## [Author Response · Author response to Decision Letter 1]

30 Oct 2024

Response to reviewers (page and line numbers refer to the manuscript showing tracked changes)

Reviewer comment 1. “Nowhere in the analysis is there any mention of the outcome based on MRI of the brain; even though there is aforementioned MRI done in three studies, we don't have their analysis of the extent of the brain injury?”

Author Response: We thank the reviewer for suggesting this aspect and we agree that the described MRI use in table 5 requires further clarification and elaboration. Although MRI was used in three studies, it was only used in a minority of patients within those studies – 27%, 56%, and 35% in the studies by Abrahams (2018) , Kali (2021) and Kali (2016) respectively. The numbers were therefore too small to draw conclusions regarding cooling efficacy. The MRIs in the studies by Kali et al. in 2021 and 2016, which were done in babies with available neurodevelopmental outcomes, were performed at widely varying times from age 7–256 and 7–145 days, with PPVs of 95% and 75% respectively.

Manuscript amendments: We have added the percentages to table 5, p16 and have added the above information to the results under, “Procedures and standards during TH in SSA studies” in lines 330-334, and to the discussion under, “Outcomes in cooled infants” in lines 518–519.

Reviewer comment 2. “In addition, studies have shown that clinical neurological examination at the end of the first month of life has predictive value regarding the outcome of newborns treated with TH at HIE. Do you have this data from your analysis? Please comment and add as this information can be very comparable to the extent that a neurological clinical analysis is performed according to a standardized methodology and a good/credible basis for comparison.”

Author Response: We thank the reviewer for commenting on this aspect since it highlights a need for further clarity which we address below. None of the studies reported neurological examination at the end of the first month of life, however this question highlighted the need give more detail for the studies which reported the use of assessments different to those described at 12 months, shown in table 9, under “other or additional assessment methods used”. None the outcomes of the methods used under 12-months were reported, but two studies reported outcomes at 18-months. The findings at 18 months were the same as 12 months in one study and in the other study there was a substantially higher loss to follow-up.

Manuscript amendments: We have added the above detail describing these aspects in table 9, p27 and in the text in results under “Hospital stays, neurological outcomes, quality of life and mortality”, on p27, lines 448–451.

Reviewer comment 3. “Do you have any data, and can you make a difference in terms of the impact of TH on reductions in death or disability, either among inborn neonates or out born neonates?”

Author Response: We welcome discussion on this important aspect. We selected mortality and disability-free survival with mild or no disability at 12 months as the most appropriate outcome indicators since these were most frequently reported. The data on survival with mild or no disability address the first part of the question, “Do you have data …of the impact of TH on reductions in death or disability”. Regarding the influence of birth site on outcome, Table 7 (p22) shows that the proportion of inborn births ranged from 33–100%, in the 11 studies with these data, but 8/11 (73%) studies included more than 78% inborn babies, and four studies did not include any out born babies. The association between birth site and outcome, was reported in two of the four birth cohort studies with adequately described cooling methods in line with ILCOR recommendations; an independent association between birth site and outcome was only shown by Damoi et al. where inborn babies were 74% less likely to die (HR: 0.26, 95% CI: 0.07, 0.94).

Manuscript amendments: We have added the above data to the results under, “Admission characteristics of babies who receive TH in SSA studies”, on p21, lines 374–380.

Reviewer comment 4. “Are there data on the outcome of children with moderate/severe hypoxia who are not treated with TH in SSA, and can these be used as a control group?”

Author Response: We agree with the reviewer that outcomes of normothermic babies with moderate-severe HIE in SSA may provide control data. Although our scoping review was limited to studies with cooled babies, seven of the included studies also described outcomes in cohorts of non-cooled babies with moderate-severe HIE. However, these cohorts should not be used as controls to

inform the efficacy of TH when applied within ILCOR guidelines due to the following reasons: In one study, they were managed in a setting without high care and ventilation facilities (Robertson 2011); in three studies the cohorts included preterm babies or did not specify gestational age (Simpson 2018, Padayachee 2013, Sebetseba 2020); in one study babies with HIE were not ventilated (Ballot 2020); in another study the criteria indicating moderate-severe HIE were not adequately described (Simbruner 1999) ; and in the final study, the majority of the non-cooled cohort consisted of babies who were not offered TH because they were deemed to be moribund (54%), too unstable for TH (10%) or too mild for TH (9%). (Nakwa 2023) Previous historical data of non-cooled babies in centers, prior to provide TH in SSA and in centers that do provide TH in SSA, were not the subject of this review, but may not offer appropriate control groups since the management and definition of HIE varies considerably with time and between centers. (Naburi 2024)

Manuscript amendments: We have added the above commentary to the discussion, under a separate subtitle, “potential control groups”, on p30–31, lines 536–557.

Reviewer comment 5. “Despite the fact that the results on mortality and intact survival rates are comparable with global norms. However, loss to follow-up was high and mortality rates were higher than in the birth cohort studies where TH or IIPPV was restricted to near-term or term babies without severe HIE, or to inborn babies only. Similarly, the CP rate was low in the survivor study where babies with severe HIE had not been admitted to NICU. Despite all of the above, can you conclude that the results are comparable to the global norm?”

Author Response: We agree with the reviewer that comparability of mortality and intact survival is compromised by exclusion of babies with severe HIE and loss to follow up after discharge in some studies. We have therefore excluded one of the studies from our conclusions, the birth cohort study by Kali et al (2015 and 2916), since 22% of babies in that cohort were cooled between 6 and 8.5 hours, which was previously overlooked as being against ILCOR recommendations. However, we also identified the previously overlooked study, a RCT comparing analgesia with TH in both groups, which was in line with ILCOR (Kali 2021), and showed 71% overall survival with mild or no disability and only 16% loss to follow up in both groups combined – this RCT and the study with aEEG data at 48-hours (Horn 2013b) both include babies from birth, and also include babies with severe HIE. The low CP rate in studies which exclude CP, does not invalidate the acceptable outcomes of the two studies with appropriate methods – it is highlighted to show potential further improvement and prompt reflection on cooling criteria.

Manuscript amendments: We have amended the following sections to indicate the departure from ILCOR by the Kali 2015 and 2016 studies due to late cooling of some infants, and to recognise the validity of the outcome data from Kali 2021: table 5 on p16; the results under, “Procedures and standards during TH in SSA studies” on p17, lines 322–323 and, “ Hospital stays, neurological outcomes, quality of life and mortality” on p25, lines 420–434; the discussion on p28, lines 471–474, p29, lines 506–507, p30, line 520 and 525–533, and p31, lines 576–581. Additionally, we have acknowledged the limitations of the other studies in the review in the conclusion of the abstract, in lines 50–52.

Reviewer comment 6. “Do you think that the effectiveness of TH treatment would be comparable to the results from the HELIX-LIC study if this analysis also included newborns with severe HIE who would need intensive treatment? Can you please comment and add in the manuscript?”

Author Response: We thank the reviewer for this question and suggestion. The two studies which included babies from birth, on which we now base our main conclusions, do include babies with severe HIE.

Manuscript amendments: The inclusion of the data from the RCT of analgesia with TH (Kali 2021) adds more robust outcomes which compare favourably to the results from the HELIX-LIC study, and we have now discussed that more clearly in lines 528–535.

Reviewer 2: No changes were recommended by reviewer 2.

We thank the reviewer for their appreciation and comments.

---

## [Decision Letter · Decision Letter 1]

21 Nov 2024

Therapeutic hypothermia for neonatal hypoxic ischaemic encephalopathy in Sub-Saharan Africa: a scoping review

PONE-D-24-28470R1

Dear Dr. Horn,

We’re pleased to inform you that your manuscript has been judged scientifically suitable for publication and will be formally accepted for publication once it meets all outstanding technical requirements.

Kind regards,

Professor Stefan Grosek, Ph.D., M.D.,

Academic Editor

PLOS ONE

Additional Editor Comments (optional):

Dear Authors

Thank you submission of you paper. The reviewers and myself agree that all issues raised by revewers were answered and we propose to accept your paper.

Kind regards

Stefan

Reviewers' comments:

Reviewer's Responses to Questions

**Comments to the Author**

1. If the authors have adequately addressed your comments raised in a previous round of review and you feel that this manuscript is now acceptable for publication, you may indicate that here to bypass the “Comments to the Author” section, enter your conflict of interest statement in the “Confidential to Editor” section, and submit your "Accept" recommendation.

Reviewer #1: All comments have been addressed

2. Is the manuscript technically sound, and do the data support the conclusions?

Reviewer #1: Yes

3. Has the statistical analysis been performed appropriately and rigorously? 

Reviewer #1: Yes

4. Have the authors made all data underlying the findings in their manuscript fully available?

Reviewer #1: Yes

5. Is the manuscript presented in an intelligible fashion and written in standard English?

Reviewer #1: Yes

6. Review Comments to the Author

Reviewer #1: Thank you for all answers.

I don't have additional questions and comments.

This manuscript satisfies the PLOS ONE criteria for publication.

7. PLOS authors have the option to publish the peer review history of their article (what does this mean? ). If published, this will include your full peer review and any attached files.

**Do you want your identity to be public for this peer review?** For information about this choice, including consent withdrawal, please see our Privacy Policy .

Reviewer #1: **Yes: ** Asist. prof. Aneta Soltirovska Šalamon

---

## [Editor Report · Acceptance letter]

PONE-D-24-28470R1

PLOS ONE

Dear Dr. Horn,

I'm pleased to inform you that your manuscript has been deemed suitable for publication in PLOS ONE. Congratulations! Your manuscript is now being handed over to our production team.

Kind regards,

on behalf of

Professor Stefan Grosek

Academic Editor

PLOS ONE